# Scoping review of the literature on outcomes of the conservation reserve program

Mark P. Nessel[1]*, Karen Maguire[2], Rich Iovanna[3], Courtney J. Duchardt[1,4], Scott R. Loss[1]

**1** Department of Natural Resource Ecology and Management, Oklahoma State University, Stillwater, OK, United States of America, **2** USDA Economic Research Service, U.S. Department of Agriculture, Kansas City, Missouri, United States of America, **3** USDA Farm Production and Conservation Mission Area, Economic and Policy Analysis Division, Washington, DC, United States of America, **4** School of Natural Resources and the Environment, University of Arizona, Tucson, Arizona

* nesselm13@gmail.com

## Abstract

The Conservation Reserve Program (CRP), one of the largest private lands conservation programs in the U.S., has played a critical role in soil erosion reduction, water quality improvement, wildlife habitat enhancement, and carbon sequestration since its inception in 1985. This paper contributes to the extensive research illustrating the benefits of the program by providing a scoping review of the peer-reviewed CRP literature. We conduct a broad-scale examination of the literature on CRP, identifying the gaps and trends in this literature to inform future research on this program. Through systematic analysis of 577 studies, we examined spatial and temporal trends in the literature, categorized CRP research into five major categories—including studies of wildlife, vegetation, air/soil/water, productivity, and social aspects—and conducted a detailed evaluation of the CRP outcomes in the wildlife and vegetation categories. For studies of wildlife-related outcomes of CRP we found studies of birds to dominate the literature, while research on other taxa such as fish, reptiles, and amphibians remains sparse. Geographically, most studies are concentrated in the Great Plains, leaving regions such as the Pacific Northwest underrepresented relative to CRP land share. This review highlights the need for long-term studies and additional research, including less-studied taxa and underrepresented regions, to better understand CRP's potential for enhancing biodiversity and ecosystem services.

## Introduction

Agriculture production is crucial for sustaining human livelihoods and societies and is also one of the most substantial ways humans alter the earth's land cover. The Food and Agriculture Organization of the United Nations estimates that agricultural land makes up 48 million km$^2$, or about 45% of all habitable land on earth [1,2], and in 2022, 45.1% of U.S. land area was estimated to be devoted to agricultural use

**Data availability statement:** All relevant data are within the manuscript and its Supporting Information files.

**Funding:** This research was funded by the U.S. Department of Agriculture (USDA) Natural Resources Conservation Service (Award # NR233A750023C009), the USDA Economic Research Service (Agreement #58-6000-2-0071), and USDA Hatch Grant funds through the Oklahoma Agricultural Experiment Station (OKL03231 and OKL02915). We thank all authors whose work was included in our scoping review. The findings and conclusions in this paper are those of the author(s) and should not be construed to represent any official USDA or U.S. Government determination or policy.

**Competing interests:** The authors have declared that no competing interests exist.

[3]. In particular, land use changes associated with agricultural crop production are contributing substantially to biodiversity loss [4,5]. Government programs designed to set aside marginal or abandoned farmland for purposes of land restoration, resource renewal, and biodiversity conservation [6–9] exist across the globe, but these programs are subject to changes in economic and political conditions, and many have only lasted for a short time without being renewed [7,10].

One of the largest and longest-standing private-lands conservation programs is the U.S. Department of Agriculture (USDA) Farm Service Agency (FSA) Conservation Reserve Program (CRP) administered under the Farm Bill Legislation. The CRP is a voluntary land conservation program that was established in 1985 with the primary goal of reducing soil erosion on environmentally sensitive cropland in the United States [11]. Subsequently, the goals have expanded to include improving water quality and providing wildlife habitat [11]. Agricultural producers in the program receive cost-share assistance for establishing approved land cover and annual rental payments for the duration of their CRP contract. Payments may also include incentive payments at sign-up or for work needed to maintain approved land cover on CRP lands. While annual enrollment fluctuates, CRP enrollment peaked in 2007 at 36.8 million acres, and in its first 35 years of existence (1986–2020) the program averaged 29.4 million acres enrolled annually [11]. Sharing a common theme of perennial cover, the broad range of practices now supported by the program encompass a wide variety of vegetation types and conservation goals. These practices, which are classified by "CP code", include "Establishment of Permanent Native Grasses" (CP2), "Riparian Buffers" (CP22), and "Habitat Buffers for Upland Birds" (CP33) [12].

Numerous studies have demonstrated a range of environmental benefits provided directly or indirectly by CRP. These benefits include providing wildlife habitat (acknowledging that habitat is a species-specific concept, we use 'wildlife habitat' as a shorthand for the variety of species-specific habitats reported in the literature) [13,14], increasing abundance and productivity of beneficial insects [15,16], fostering floral diversity [16,17], mitigating flood risk [18], restoring hydrology and reducing nutrient runoff [19,20], enhancing groundwater recharge [21], reducing soil erosion [22], and storing carbon and reducing greenhouse gas emissions [23,24]. Beyond these environmental benefits, CRP also provides consistent annual rent payments to farmers, which may benefit producers during periods of market decline or who seek to dampen the effects of price and yield variability [25,26].

Limited time and resources mean that most ecological studies of the environmental benefits of CRP have been conducted by a single team of researchers working in a local study area, often for only a few years. However, CRP contract goals and implementation vary across the diverse ecosystems of the United States. Examining and describing CRP benefits broadly, across types of CRP and regions, may help inform potential program participants and policymakers about the benefits of CRP for land conservation. A broad-scale view of CRP allows us to identify trends across taxa, regions, ecosystem types, and CRP types, and to identify gaps in research to date.

Here, we conduct the first scoping review of peer-reviewed CRP research. A national-level systematic scoping review allows us to identify major research emphases and gaps in the literature. We discuss the specific outcomes, regions, taxa and CRP program characteristics that have been well-studied and highlight those that may require further research. Specifically, we systematically evaluated CRP literature to address three main objectives: 1) characterize spatial and temporal trends in CRP research, 2) categorize CRP research by type, including studies of wildlife; vegetation; air, soil, and water; and more human-focused aspects (e.g., economic benefits and stakeholder perceptions of CRP), and 3) evaluate detailed characteristics of studies that focus on responses of wildlife and vegetation to CRP. With this scoping review, we seek to inform and guide future research on important aspects of CRP that have to this point not received formal empirical evaluations in the literature.

## Methods

### Search & exclusion criteria

We searched scientific literature sources for all studies that pertained to the Conservation Reserve Program. Using the search term "Conservation Reserve Program" (search conducted on February 15, 2024) we collected all studies on Web of Science (n = 966) and Scopus (n = 940). The two searches were combined, and duplicate studies were removed (n = 1167 unique studies; S1 Table). Our review focused on studies specifically examining CRP outcomes and responses, as well as perceptions and behaviors related to CRP (hereafter "CRP studies"). Studies could have directly compared CRP to other land cover types, compared different CRP practices, or measured a response that included CRP as a factor. Our search included only peer-reviewed journal articles and excluded government reports that were not indexed in Web of Science or Scopus, article types that are either not peer-reviewed or were unlikely to contain original information about effects of CRP (i.e., editorial materials, meeting abstracts, theses, commentaries). Further, in our preliminary search we found we were not able to comprehensively capture such non-peer-reviewed sources using our databases. Using broader sources like Google Scholar did allow us to capture non-peer-reviewed literature, but also exponentially increased the number of sources to sort through, making this method impractical. We also excluded studies that included data collected at CRP sites but did not evaluate CRP *per se* (e.g., using previously collected CRP data to test the efficacy of a tool or model without drawing inferences about outcomes associated with CRP).

The literature search retrieved 48 review papers examining the effect of CRP but putting forward no novel data. These review papers, which ranged from systematic meta-analyses to literature reviews specific to a particular topic, were not included in our study counts as the data they contained would be captured in other studies collected in our database. We could not access a small number of papers retrieved in our search (n = 43), and information from the title, abstract, and keywords of these papers was not sufficient to include them in our final database. These studies were either published in smaller, trade specific publications that we were not able to access or were published prior to 2000 making it possible that they do not exist in a viewable form online.

### Study categorization

After filtering out unsuitable papers, the remaining studies (n = 577; S1 Table) were broadly categorized into 5 major groups: Vegetation, Wildlife, Air/Soil/Water, Social and Productivity. The first three groups include studies measuring the effect of CRP on non-crop plants, animals, and air, soil and/or water respectively. We chose to combine papers examining air, soil and water as many of the papers we collected measured similar responses (e.g., carbon content, nutrient content) in more than one of these categories, and in some cases, responses included aspects of more than one of these components (e.g., sediment runoff involves both soil and water). The Productivity category included studies examining land use or agricultural productivity, the latter of which could be measured in either dollars or crop biomass. The Social category included papers studying the behavioral responses associated with CRP contracts (e.g., producer choices in CRP

auctions), and other individual and community effects of CRP. These five categories were not mutually exclusive, and one study could be eligible for multiple categorizations depending on its scope. For consistency and to avoid editorializing the findings that were evaluated, we based categorizations on the primary results presented in the study. For example, plant height can be both a measure of wildlife habitat and plant physiology. If a study presented results as "plant height," we would consider it to be a vegetation study, whereas if the same measure was presented as habitat structure or habitat quality, we would consider it to be a wildlife study.

To address the objectives of this scoping review, we generated total counts of studies and calculated proportions of studies in different categories in order to identify trends and gaps in the literature. Because this is a scoping review and not a meta-analysis or narrative review, and because our objective was to characterize trends, emphases, and gaps in what has and has not been studied with regard to CRP effects, we intentionally did not extract or synthesize the results from our studies, and instead only focused on what the literature did and did not evaluate. For each eligible study, we extracted the following data if available: location data, including study site, and state, the duration of the study and season when data were collected, and CRP-specific information such as the number of CRP sites studied, Conservation Practice (e.g., CP1, CP2, etc.), and the number of years sites were enrolled in the program. As with the study categories above, many of these categories were not mutually exclusive, meaning one study could contribute multiple data points (e.g., a study that evaluated multiple conservation practices). However, because such studies were only included in descriptive summaries of total counts of studies or to calculate proportions of total studies (e.g., X% of studies examining bird species), any individual study was only counted once for any single metric that we presented. We compared the final number of studies conducted in each state to total CRP acreage [27] to determine if more CRP studies have been conducted in states with larger shares of CRP land. The amount of land enrolled in the program fluctuates by year, so data from 2017 was chosen as it was the year after the greatest number of CRP studies meeting our study inclusion criteria was published. For Vegetation and Wildlife studies, we also noted the types of reported results, which we grouped into general categories for further classification (Table 1).

Given the threat to biodiversity from land cover change noted earlier, we also collected more detailed information about CRP studies categorized as either Vegetation or Wildlife. Specifically, we noted if particular species were reported in the results. We used scientific names for each organism if provided. If only common names were provided, then the scientific name was retrieved and used. All names were checked and updated to present-day nomenclature. We also recorded whether vegetation studies made comparisons that we considered to be especially relevant ecologically, including comparisons of warm season (C4) vs. cool season (C3) grasses, and native vs. non-native plants (all specific comparisons in Table 2). For these vegetation comparisons, we were only interested in CRP treatments, so non-CRP plots included in studies were not included in these comparisons.

The majority of wildlife studies in our database focused on birds. To assess conservation status of bird species that have been studied in the context of CRP, we utilized the Avian Conservation Assessment Database (ACAD) [28]. These scores designate species of greatest conservation concern due to small and/or rapidly declining populations, limited distributions, and high threats throughout their ranges. Species in the Red Watchlist are of highest concern, followed by Orange then Yellow. A full explanation for how each color coding is scored is available on the ACAD website [28].

## Results

### Spatial and temporal trends in CRP research

Of the 577 studies collected in our database the largest group was Air/Soil/Water (188) followed by Wildlife (169), Productivity (162), Vegetation (77) and Social (56) (Fig 1). The earliest study captured in our search that contained the phrase "Conservation Reserve Program" was published in 1985, but it did not include an examination of CRP effects as the program was established that year. The first studies explicitly evaluating CRP effects are from 1987, including studies

**Table 1. Vegetation and Wildlife results reported by studies.**

**Vegetation**

| Name | Description | Examples |
|---|---|---|
| Biomass | Measure of plant or plant tissue to establish the amount in a given space | Biomass, Abundance |
| Cover | Measurement of space on land taken up by plant or plant type | Percent cover, percent grass |
| Diversity | Measure of number or composition of plant species | Diversity, Evenness, Species composition |
| Disease | Measure of disease severity or plant response to disease severity | Number of spots, pathogen cell counts |
| Physiology | Measure of plant traits at individual plant level from cellular to whole plant | Growth rate, Number of leaves |
| Reproduction | Measure of seed/fruit production and biomass | Number of seeds, seed size, fruit size |
| Structure | Measure of height taken up by vegetation | VOR, vegetation height |

**Wildlife**

| Name | Description | Examples |
|---|---|---|
| Abundance | Measure of number or biomass of wildlife | Biomass, Abundance |
| Diet | Measure of amount or richness of food being utilized by wildlife | Food diversity, Foraging Rates |
| Distribution | Measure of detection, movement or occurrence of wildlife | Habitat Use, Detection Probability, Occupancy |
| Diversity | Measure of number or composition of wildlife species | Diversity, Evenness, Species richness |
| Ecological Service | Measure of specific ecosystem services provided by the wildlife being studied | Avian conservation value, Honey Yield, Pollination |
| Mortality | Measure of mortality or predation rate of wildlife | Mortality, Predator rates, Survival |
| Reproduction | Measure of number and survival of wildlife broods | Brood Count, Fecundity, Nest Success |

**Table 2. Types and number of comparisons made in vegetation studies.**

| Conservation Practices | # Studies | Functional Groups | # Studies | Management | # Studies |
|---|---|---|---|---|---|
| CRP Age | 20 | Cool-warm/C3-C4 grasses | 8 | Grazed | 9 |
| CRP Type | 12 | Native-Non-Native | 3 | Herbicide | 9 |
| | | C4-other | 1 | Hayed | 7 |
| | | Interseeding | 1 | Mowed | 6 |
| | | Old World bluestem-native mixed species | 1 | Burned | 5 |
| | | Seral-Native | 1 | Disc | 5 |
| | | | | Fertilizer | 3 |
| | | | | Harvested | 3 |
| | | | | Previous crops | 1 |
| | | | | Tillage | 1 |
| | | | | Wooded | 1 |

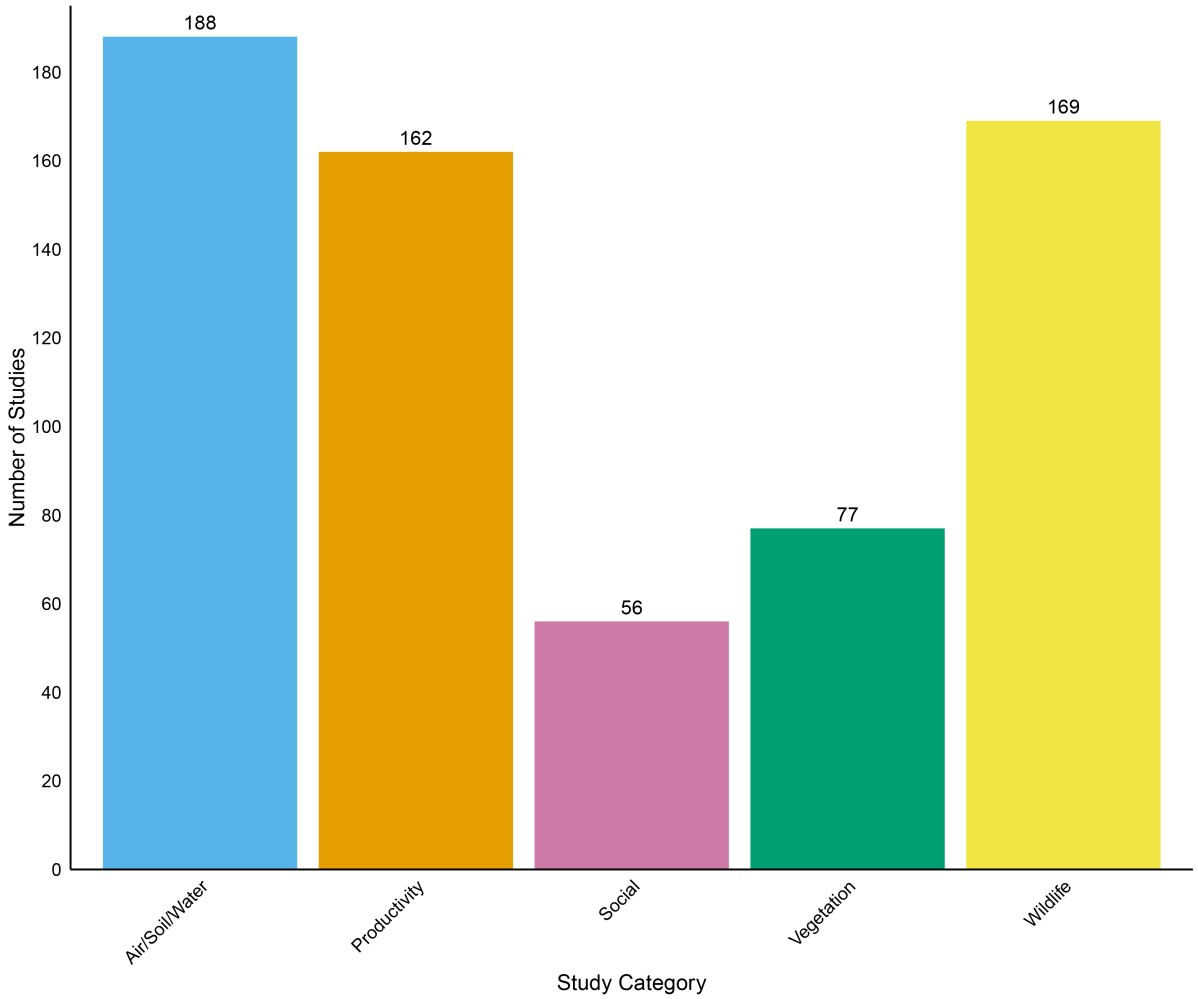

**Fig 1. Number of studies from each category collected in our database.** Some studies are counted more than once in this graph because they fall under multiple study categories.

modeling agricultural productivity and program payouts, as well as studies examining CRP effects on air, soil and water quality (Fig 2). In 1990 the first studies examining the effect of CRP on farmer decisions as well as public opinion of CRP (Social) were published (Fig 2). Studies examining the effects of CRP on vegetation and wildlife began to be published in 1993 (Fig 2). During the 30-year period between 1994 and 2023 an average of 20 studies examining the effects of CRP were published per year ranging from 11 studies in 1995–36 studies in 2016.

Regarding the spatial distribution of CRP research, Kansas has had the most published studies overall (Fig 3) as well as the most vegetation and wildlife studies (Fig 4). Kansas and North Dakota both had the most Productivity studies (Fig 4). Illinois has hosted the most Social studies of any state, while Nebraska has had the most studies on Air, Soil and Water (Fig 4). The largest number of studies have occurred in states in the central part of the U.S.; specifically, 46% (n = 230) of studies were in states along the 98th meridian (i.e., North Dakota, South Dakota, Nebraska, Kansas, Oklahoma and Texas), with most of the studies in our dataset occurring in Great Plains states (Fig 3). When comparing this spatial distribution of studies to CRP enrollment in 2017 (S2 Table), the year after the greatest number of CRP studies meeting our study inclusion criteria were published, states with high CRP acreage tended to also have high numbers of CRP

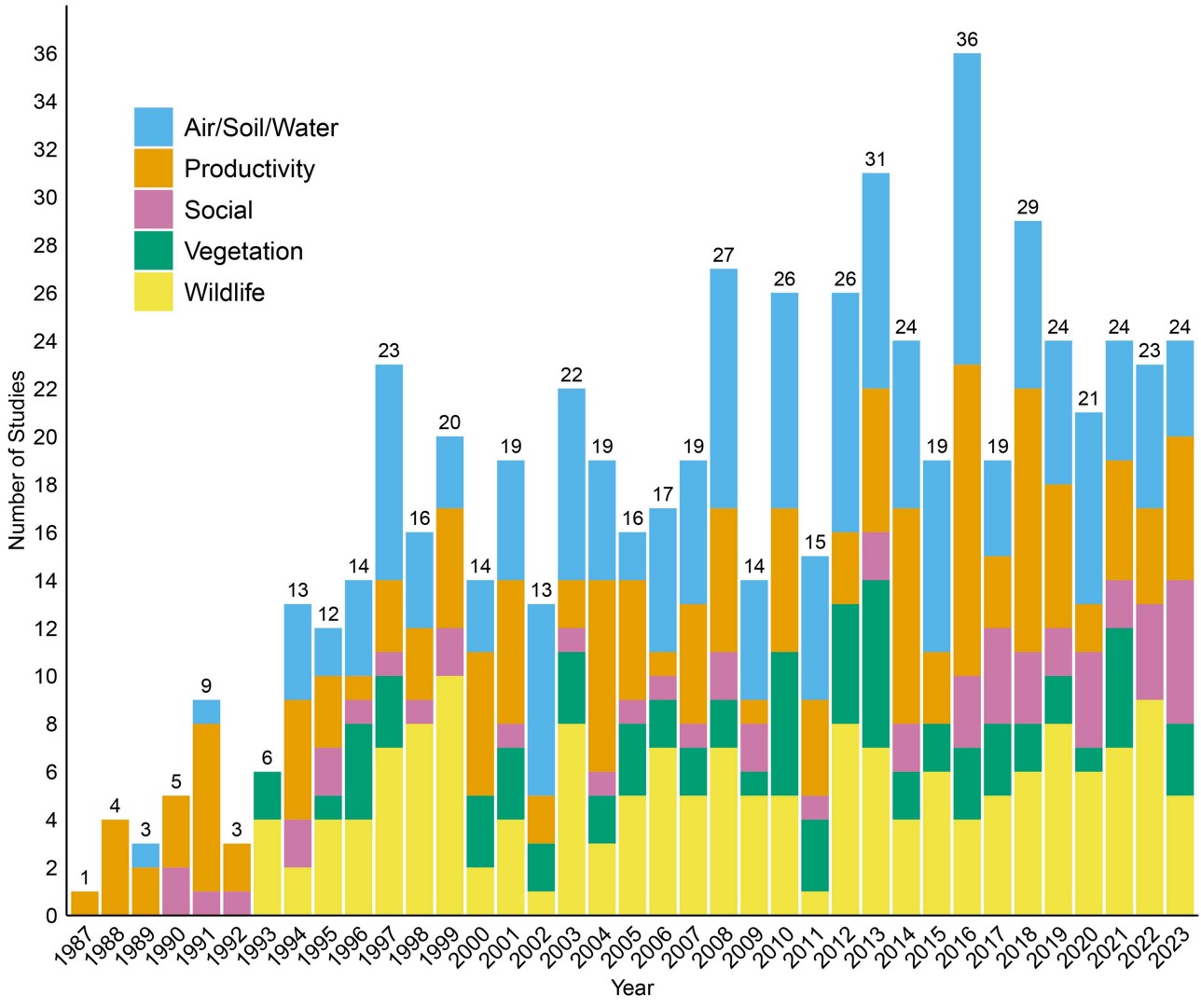

**Fig 2. Number of studies from each category collected in our database separated by publication year.** Some studies are counted more than once in this graph because they fall under multiple study categories. Studies from 2024 are not well represented as our literature search began in early February 2024; therefore, the 2 studies we collected that were published in 2024 were excluded from the figure.

studies. The top 10 states in terms of enrollment averaged 54 studies each; 64% (n = 320) of all studies that included a specified location occurred at least in part in these 10 states. The most obvious exception to this pattern is Washington, which had fewer CRP studies than expected based on enrollment; this state ranked seventh in terms of CRP acreage but 16th in number of studies with 26 total. Conversely, Wisconsin and Michigan had more studies than expected based on their share of enrollment, ranking 24th and 30th respectively in terms of CRP acreage but 14th and 13th in number of studies (S2 Table). Our database contained 82 studies that were described as using "National" level data, meaning the CRP data in the study was likely collected from all or most states with land enrolled in CRP. However, the specific states represented in the data were not specified in these national studies, preventing inclusion in our state counts. Most national studies (50 studies; 62%) were in the Productivity category, typically utilizing nationwide surveys or satellite images of land cover, 21 studies (26%) were in the Air/Soil/Water, 13 were categorized as Social (16%), and 8 (10%) were categorized as Wildlife

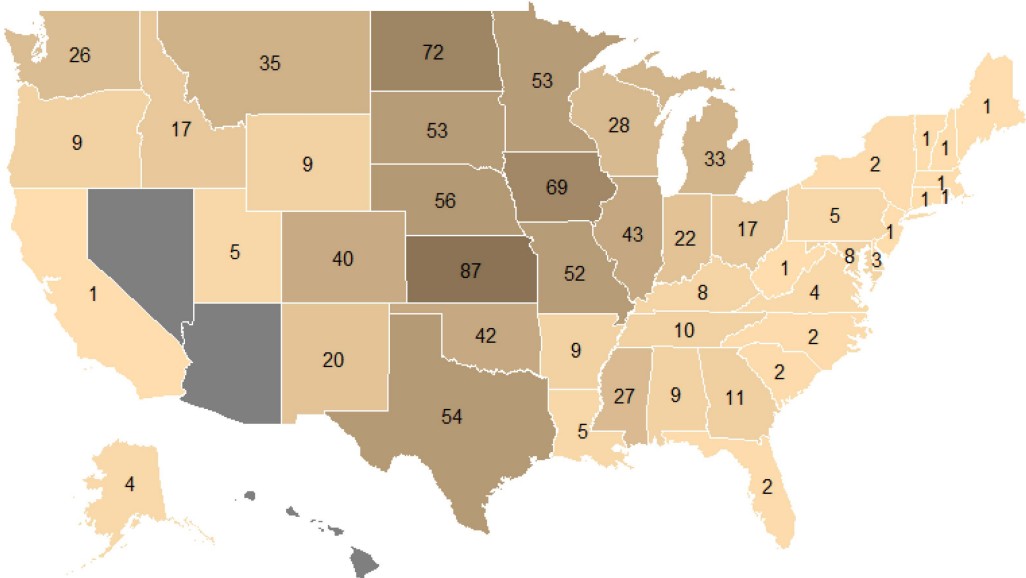

**Fig 3. Map depicting total number of studies of CRP effects that occurred in each state.** This does not include 85 studies that included "National" level data, which did not specify what state or states the data was collected in.

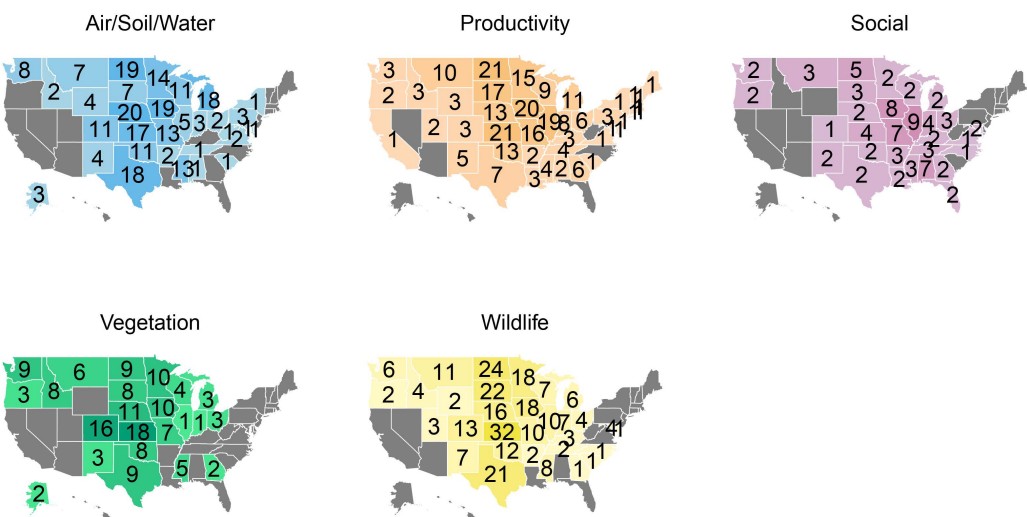

**Fig 4. Map depicting number of studies that occurred in each state, separated by study category.** This does not include 85 studies that included "National" level data, which did not specify what state or states the data was collected in.

with no national scale Vegetation studies. We did not find any studies pertaining to CRP for three states (Arizona, Nevada and Hawaii) or for Puerto Rico, which also participates in the program (Fig 3).

## Vegetation and wildlife responses to CRP

We found 77 studies that examined vegetation including measurements of grasses, forbs, trees, shrubs, and mixed communities capturing more than one of these plant types. However, only 14 of these studies specified the plant group

evaluated (e.g., grass, tree) and only 9 were explicit about the species or genus of interest (e.g., *Asclepias spp.*, *Panicum virgatum*, *Pinus palustris*), so we were unable to further examine trends in types of vegetation studied. Finally, we retrieved 169 studies that measured CRP effects on wildlife (e.g., effects on abundance, behavior, or reproduction) or wildlife habitat (e.g., habitat use or quality). All wildlife studies were explicit about the organism(s) of interest (i.e., bird, fish, invertebrate, mammal or reptile), and wildlife research covered a wide range of topics including population dynamics, behavior, and ecosystem services pertaining to the species of interest (e.g., pollination, pest removal).

Of the studies in wildlife and vegetation groups, 75 vegetation studies (97%) and 151 wildlife studies (89%) reported study duration. Vegetation studies ranged from 1 to 9 years in duration (average = 2 years, median = 2 years), while wildlife studies ranged from 0.5 to 23 years (average = 2.5 years, median = 2 years). For both groups, the most common study duration was one year (30 of 75 vegetation and 20 of 151 wildlife studies).

Among 77 studies that included descriptions of the specific vegetation variables assessed in the context of CRP, most (n = 54) included vegetation cover measurements (Fig 5). Most studies (n = 68) did not explicitly examine CRP effects on a particular species, but the few that did evaluated either invasive species (e.g., *Ventenata dubia*), species of conservation concern (e.g., *Pinus palustris*), or species useful for agriculture (e.g., *Panicum virgatum*). Regarding vegetation comparisons, 23 studies (29%) did not include vegetation comparisons in our previously identified categories deemed ecologically important (Table 2). Studies that did include such vegetation comparisons fell into 3 broad categories, including comparing outcomes for: fields enrolled in different periods or different CRP practices; plants in different functional groups (e.g., C3 to C4 grasses, native to non-native species); or different management approaches (e.g., hayed, disced, burned) (Table 2).

The majority of wildlife studies (n = 107) evaluated how CRP was associated with some measure of animal abundance (e.g., average counts, biomass; Table 1, Fig 6). The next most common measures were reproduction (n = 42), diversity (n = 37), and distribution (n = 27) (Table 1, Fig 6). We found very few studies focused on wildlife ecosystem services (n = 9), mortality (n = 6), or diet (n = 5). Nearly all wildlife studies pertained to birds (n = 115; Fig 7). The next most-frequently studied taxa were invertebrates (n = 36); most invertebrate studies focused on insects, but some evaluated other terrestrial invertebrates such as arachnids and annelids (Fig 7). We also collected 18 studies on mammals, which tended to focus on deer, coyote, and rodents like prairie dogs and prairie mice (Fig 7). We only found a single study focused on fish or reptiles (Fig 7) and did not find any studies examining other important wildlife groups such as amphibians as well as important mammal groups like bats.

Among the 115 studies examining birds, 160 unique species were studied (S3 Table); of these, 55 species were only reported in a single study. The majority of the bird species studied were in the order Passeriformes (songbirds; 104 species) followed by Charadriiformes (shorebirds like sandpipers and plovers; 11 species), Galliformes (e.g., turkey, pheasant, and grouse; 10 species) and Anseriformes (waterfowl; 9 species). Grasshopper Sparrow (*Ammodramus savannarum*) was the most studied species (50 studies), followed by Dickcissel (*Spiza americana*; 41 studies) and Ring-necked Pheasant (*Phasianus colchicus*; 38 studies) (Table 3). When considering ACAD's watchlist, only 25 species (16% of those studied) were on the list, including 9 species on the Red Watch List, 4 on the Orange Watch List and 12 on the Yellow Watch List (S3 Table). The three most frequently studied species with a watch list designation were Bobolink (*Dolichonyx oryzivorus*; Orange Watch List; 28 studies), Field Sparrow (*Spizella pusilla*; Yellow Watch List; 20 studies), and Henslow's Sparrow (*Centronyx henslowii*; Yellow Watch List; 15 studies). Lesser Prairie-Chicken (*Tympanuchus pallidicinctus*; 12 studies) was the most commonly studied Red Watch List species. The 25 species on ACAD's watchlist appeared in 5 studies each on average (range 1–28) with 9 of the species appearing in only a single study (S3 Table).

## Discussion

We conducted the first scoping review of peer-reviewed journal publications of CRP research with the goal of quantifying trends in the literature and identifying potential gaps in the research on this long-standing private lands conservation program. Our analysis uncovered several trends: 1) The published research on CRP has been relatively consistent

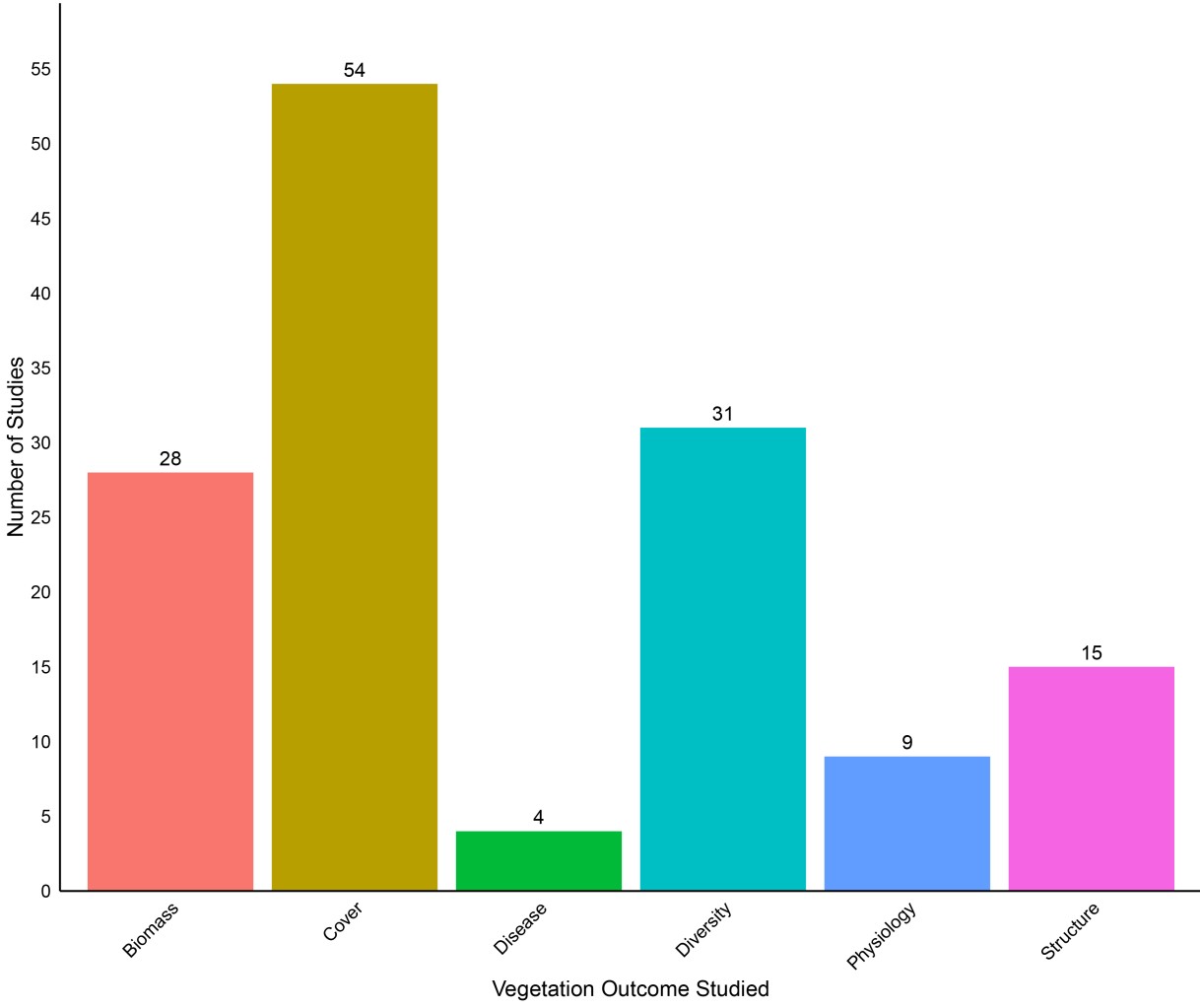

**Fig 5. Number of Vegetation studies from each category collected in our database.** Some studies are counted more than once in this graph because they fall under multiple study categories.

temporally despite declines in enrollment, and the majority of research has been conducted in regions that have had the greatest share of CRP enrollment (the Great Plains and upper Midwest), 2) The largest share of the research has pertained to CRP's effect on air, soil and water, and vegetation studies are the least common, 3) The focus of vegetation and wildlife studies has largely been on vegetation cover and wildlife abundance and the vast majority of wildlife-related CRP studies have evaluated bird responses to the program, with a marked lack of studies on fish, reptiles, and amphibians.

## Spatial and temporal trends in CRP research

There were no major year-over-year changes in the type or number of studies published about CRP since the program's inception in 1985. Beginning in 1994 we observed no multi-year trends in the number of published studies in all 5 categories. Although CRP enrollment has declined by more than 10 million acres since 2007 [11], we did not observe a concomitant decline in published CRP research. These consistent rates of publication concurrent to declining CRP enrollment may reflect trends in wildlife-focused research, and specifically, an increase of this research on existing CRP-enrolled

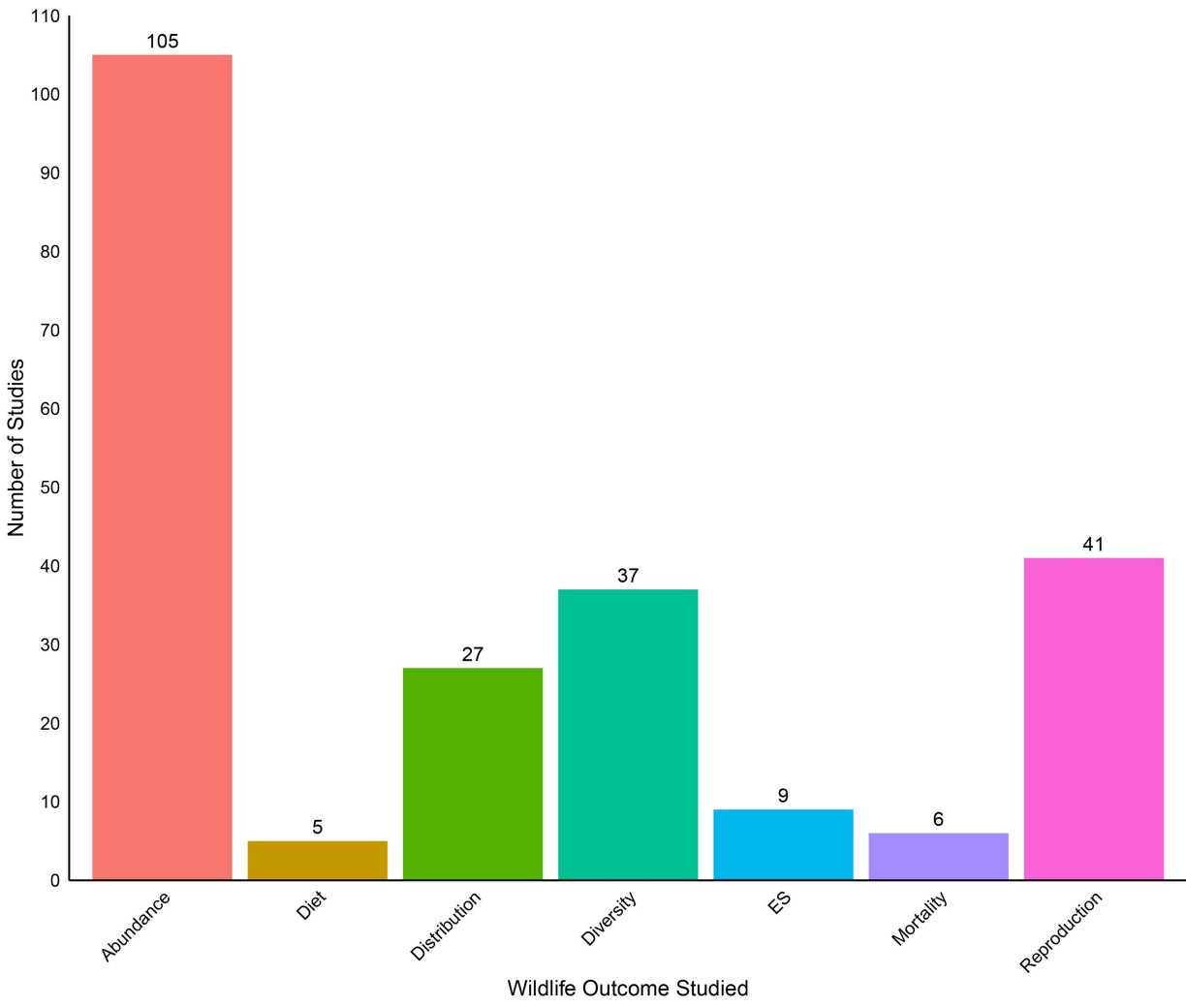

**Fig 6. Number of Wildlife studies from each category collected in our database.** Some studies are counted more than once in this graph because they fall under multiple study categories.

land toward taxa that were previously understudied, both on CRP lands and more broadly. For example, prior to 2013 only a single paper published in our database [29] focused on insects outside the context of plant pests or vertebrate prey. Starting in 2013 these insect papers began to capture a much wider gamut of topics including insect community structure, biodiversity, and pollinator nutrition; with only 3 papers from this time period primarily covering insects as prey or pests. Perhaps the larger number of studies on the effects of CRP on insects reflects recently shifting priorities concerning the importance of invertebrates, especially of the ecosystem functions and services provided by pollinators [30,31].

Our scoping review uncovered clear spatial patterns in CRP studies, with most conducted in states on or near the 98th meridian (Figs 3,4). These states also tend to have the most CRP land; specifically, the 6 most-frequently studied states (North Dakota, South Dakota, Nebraska, Kansas, Oklahoma and Texas) are in the top 15 in terms of CRP land acreage (S2 Table), and surrounding states like Iowa, Montana and Minnesota also have relatively high CRP acreage and numbers of studies (S2 Table). However, not all states followed this trend. For example, some states in the Great Lakes region (Wisconsin and Michigan) had a high number of studies relative to their CRP enrollment, and Pacific Northwest States

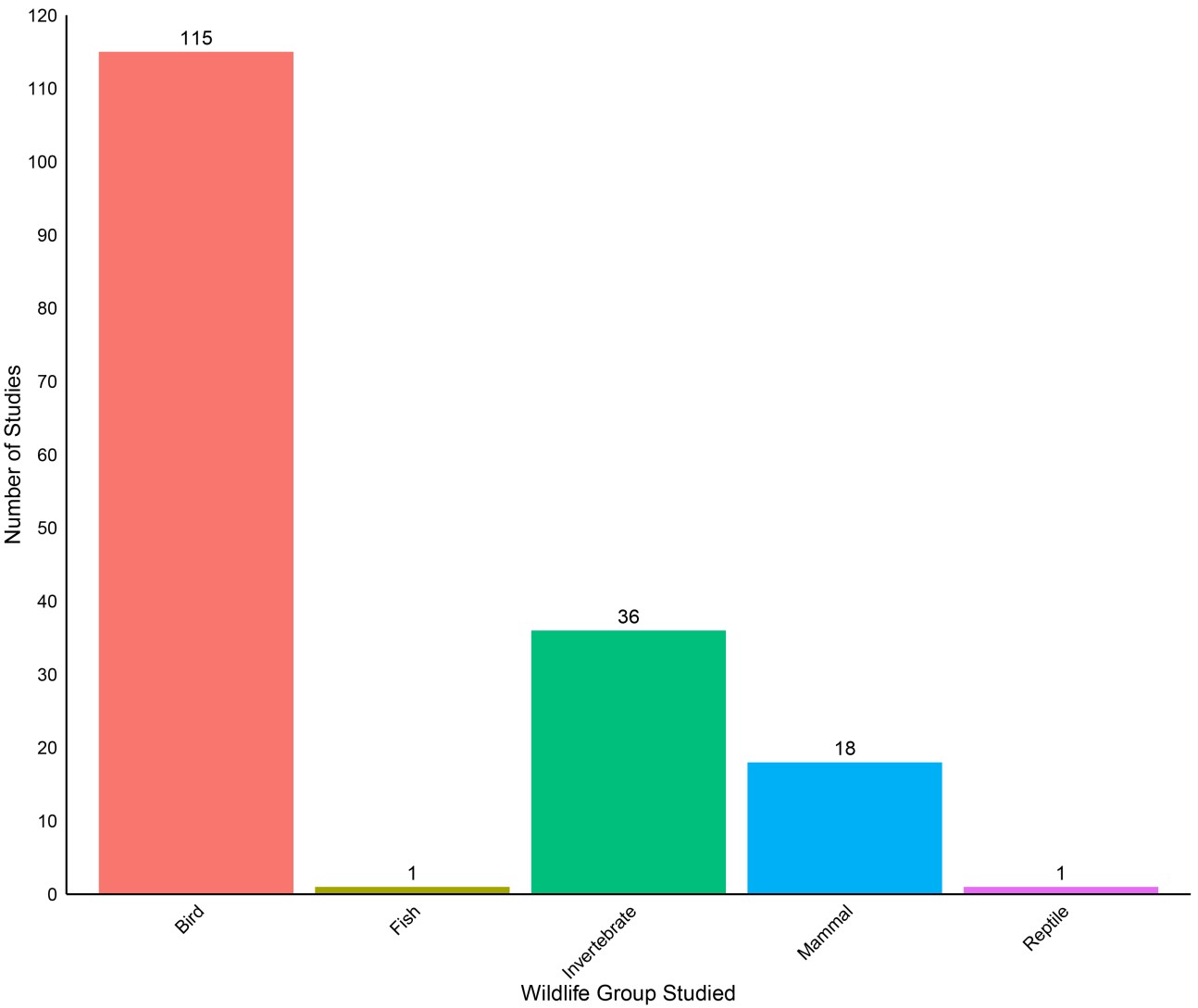

**Fig 7. Number of Wildlife studies examining major animal groups.** Some studies are counted more than once in this graph because they examined more than one type of animal.

(Washington and Oregon) had relatively few studies relative to enrollment. Identifying the factors driving these regional patterns is beyond the scope of this study. However, these spatial trends may relate to two key factors. First, the number of studies is likely related to the population of researchers available to study CRP in universities and other research institutions within each state. Second, CRP lands often represent a relatively large share of remaining grassland cover in heavily cultivated regions such as the Corn Belt and Great Plains, making CRP especially important for supporting habitat for various wildlife species and a focal point for conservation research. Despite the wealth of research presented herein, underrepresentation in some states may be problematic from a management perspective. Species and ecosystems respond not only to land management but also to interactions between land management and underlying topoedaphic and climatic regimes [32–34] For example, the Grasshopper Sparrow, which we found to be the most frequently studied bird in the context of CRP, has been shown to respond positively to disturbances like fire and grazing at the eastern edge of

**Table 3. Names and taxonomic information of the 10 most commonly studied birds.**

| Species | Common Name | Family | Order | # Studies |
|---|---|---|---|---|
| *Ammodramus savannarum* | Grasshopper Sparrow | Passerellidae | Passeriformes | 50 |
| *Spiza americana* | Dickcissel | Cardinalidae | Passeriformes | 41 |
| *Phasianus colchicus* | Ring-Necked Pheasant | Phasianidae | Galliformes | 38 |
| *Agelaius phoeniceus* | Red-Winged Blackbird | Icteridae | Passeriformes | 35 |
| *Molothrus ater* | Brown-Headed Cowbird | Icteridae | Passeriformes | 35 |
| *Zenaida macroura* | Mourning Dove | Columbidae | Columbiformes | 31 |
| *Sturnella magna* | Eastern Meadowlark | Icteridae | Passeriformes | 31 |
| *Geothlypis trichas* | Common Yellowthroat | Parulidae | Passeriformes | 30 |
| *Dolichonyx oryzivorus* | Bobolink | Icteridae | Passeriformes | 28 |
| *Sturnella neglecta* | Western Meadowlark | Icteridae | Passeriformes | 27 |

its range, but to respond negatively to these disturbances at the western edge of its range [35]. Thus, studies are needed that evaluate the degree to which CRP provides the required habitat conditions for such species across their entire ranges. The Pacific Northwest and other understudied regions differ ecologically and culturally from regions where most CRP research has occurred, and the nature of CRP benefits are similarly expected to be regionally variable. As such, additional research may be necessary in underrepresented regions or for underrepresented outcomes, especially as CRP enrollment and practices change over time.

## CRP studies of Air/Soil/Water, productivity, and social aspects

It is perhaps not surprising that one of the largest categories of CRP research was "Productivity", as we defined many broad and important study subjects to fall under this category (e.g., agricultural productivity, land use, economic output). While CRP is a land cover restoration program, studies are still needed that examine farm-level responses to program enrollment, as well as behavioral responses of producers to changes in the program. We found 56 studies using surveys and interviews to understand how stakeholders respond to CRP, including their perceptions of the program and the factors influencing whether landowners are likely to enroll in or stay enrolled in the program. Quantifying the value of CRP to individuals and communities often involves surveys to elicit information on the value of the environmental benefits it provides. Measuring and articulating the benefits is necessary for adaptive management within the program, and for improving outcomes for farmers, local communities, and the environment [36–38]. This body of research, including studies of the human dimensions of CRP, has underscored the broader challenge of balancing conservation and land restoration goals of CRP with the other factors that influence stakeholder decisions about land use in agricultural landscapes [39].

Reducing soil erosion and improving local environmental quality are major goals of the CRP program, and in alignment with those goals, we found that the largest group of studies (35%) in our database pertained to the effect of CRP on a combination of air, soil, and water (Fig 1). Since the program's inception many studies have confirmed that CRP adoption reduces air pollution and greenhouse gas emissions [40–42], soil erosion [43] and water degradation [44], and also provides flood risk mitigation [45] and increased carbon retention [46]. Carbon in particular seemed to be a major focus of these studies, with papers examining how carbon retention in CRP-enrolled lands is affected by soil and water use [47–49], land use [50], and productivity [51,52].

## Vegetation and Wildlife Responses to CRP

Restoring wildlife habitat is also an important CRP goal, and studies of vegetation and wildlife outcomes were well represented in our database, with 77 vegetation and 169 wildlife studies. Vegetation studies were found to be the least represented of

the 5 broad categories. This may be because fostering vegetation (e.g., diversity, cover) per se is not a major goal of CRP, and instead is a means to achieve outcomes that are captured in our other study categories (e.g., reduction of soil erosion, enhancement of wildlife habitat). However, in recent years, increasing understanding of the role of plant functional groups, or even of specific plant species, in providing such ecosystem services, is leading to a greater interest in vegetation-focused outcomes. The current iteration of CRP now includes several conservation practices that encompass land restoration activities in multiple vegetation and land cover types including wetlands, riparian areas, forests and shrublands. Although studies of effects of CRP on these cover types exist among the papers in our dataset, most research still appears to focus primarily on grasslands, and to compare CRP to other land uses, especially agricultural uses. Research evaluating how a broad range of ecosystem types respond to surrounding CRP lands would provide insight into how these CRP vegetation communities are responding to ongoing anthropogenic changes such as nutrient enrichment or climate change.

Wildlife in particular have become a larger priority in the CRP program over time (https://www.nrcs.usda.gov/ceap/wildlife), as reflected by the large proportion of CRP studies focused on wildlife outcomes, and by establishment of newer CRP practices, starting in the mid-2000s, that allow stakeholders to directly curate lands to benefit ecologically important wildlife species (CP38) including beneficial birds (CP33, CP37) and pollinators (CP42) [12]. CRP supports wildlife through increasing habitat area and quality [53,54], which in turn generates a number of ecosystem services for humans. For example, the third most-commonly studied bird species in our dataset was a game species (ring-necked pheasant), and other game species included greater prairie-chicken, northern bobwhite, white-tailed deer, wild turkey, and several duck species. Game species provide meat and sport for millions of stakeholders and, through hunting licenses, provide funding for wildlife conservation through the Pittman-Robertson Act [55]. Game and non-game species alike also provide numerous ecosystem services that benefit the environment and society, including pest removal, pollination, and wildlife viewing [56,57]. Our review illustrates that research has addressed effects of CRP on a wide variety of game and non-game wildlife species; however, we found few wildlife and vegetation studies that directly measured ecosystem services, and those that did mainly examined insect pollinators (see further discussion of pollinators below). Further research that continues to evaluate the wildlife and vegetation-mediated ecosystem services provided by CRP, especially how these services are affected by implementation of programmatic changes will be crucial to further increasing understanding of the role of CRP in providing benefits to the environment and humans.

The vegetation and wildlife research in our database was composed mostly of short-term studies, with vegetation studies averaging 2 years in duration, wildlife studies averaging 2.5 years, and with the largest number of studies in both categories focused on a single year. This short-term study duration is expected as funding allotments and job appointments (e.g., graduate student positions) tend to only cover increments of a few years. Although short-term studies can provide invaluable insights about associations between CRP and outcomes like wildlife and vegetation abundance, occurrence, species diversity, and reproductive rates, long-term and large-scale studies are important for accurately measuring ecological trends and capturing lag effects that might not manifest until years after management is applied [58,59]. Further, long-term studies, despite being rarer, are disproportionately cited in the literature and by policy makers [60]. Despite the challenges inherent in long-term research [61], such efforts are warranted in the context of CRP. Ecological restoration, which is a key desired outcome of CRP from a wildlife habitat perspective, is an inherently dynamic process [62,63]. Assumptions about predictability of land restoration (Carbon Copy Myth [62]) or immediate responses of wildlife to newly-restored habitat (Field of Dreams Myth [62]) may not be met, and it may take time to observe these responses in vegetation and wildlife. Further, the longer-term ecological studies that have been conducted sometimes reveal different, or even opposing outcomes relative to shorter-term studies [64]. Thus, care should be taken in interpreting short-term snapshot studies, and CRP studies over a longer duration could examine the dynamic effects of CRP (e.g., evaluate outcomes relative to time since CRP enrollment).

We noted several additional gaps and emphases in wildlife-focused CRP studies that may also help to inform future research. Most Wildlife studies focused on measures of wildlife abundance like density, abundance, or biomass of all

individuals (Fig 6). Metrics related to abundance and density are the main currency of conservation and population management activities, and the center of many methodological discussions [65,66], but using abundance alone to inform management decisions can in some cases be misleading. For example, if animals are drawn to habitats that decrease survival [67], abundance may be a confounding indicator of habitat quality and success of land management activities, including those associated with CRP. Further, underlying drivers of behavior [68] or habitat quality [69] may obscure interpretations of abundance trends. Researchers should balance the focus on local abundance with integration of outcomes that more directly capture population trends and indicators of community health (i.e., reproductive success). Alternatively, a more holistic understanding of large-scale abundance trends in response to CRP and other land management approaches would ensure that abundance metrics more adequately capture overall population status in response to management.

We also found biases in the types of wildlife studied in CRP publications, with birds being included in the majority of studies (68%) (Fig 7). This emphasis on birds has been shown to exist throughout the ecological literature [70], and likely reflects societal and funding agency interests, ease of monitoring, as well as the diverse ecosystem services provided by birds, including pest control, seed dispersal, pollination, and aesthetic value [71]. However, there is clearly a need to evaluate CRP effects on other, lesser-studied organisms. Amphibians were not studied in any of the publications retrieved in our study, although we are aware of at least one amphibian study not retrieved using our formal search terms [72]. Amphibians have declined in recent years for several reasons including habitat degradation associated with agricultural intensification [73,74]. Additional research focusing on understudied wildlife groups can provide a more complete understanding of how CRP is affecting wildlife communities.

North American birds, including many species of grassland-dependent birds, are declining at alarming rates [75], and this may be an additional reason for the focus of CRP research on birds. Of the 160 bird species observed in our database, 135 (85%) are not listed under any of the ACAD watchlist designations (i.e., the Red, Orange, or Yellow Watchlists). Of the 10 most-frequently studied bird species, only Bobolink (Dolichonyx oryzivorus) had a watchlist designation. Of the remaining Watchlist birds that have been studied, only 6 had been studied in more than 5 instances: Field sparrow (Spizella pusilla), Henslow's sparrow (Centronyx henslowii), Lesser prairie-chicken (Tympanuchus pallidicinctus), Chestnut-collared longspur (Calcarius ornatus), Baird's sparrow (Centronyx bairdii), and Greater prairie-chicken (Tympanuchus cupido). Some of these species, particularly Field Sparrow, Henslow's Sparrow, Lesser Prairie-Chicken, and Greater Prairie-Chicken are believed to have benefitted substantially from CRP, with steep declines through much of the 20th century appearing to have stabilized to some degree in part due to efforts to restore their habitat [13,14]. For other species, such as Baird's Sparrow and Chestnut-collared Longspur, CRP may offer less suitable habitat at the site scale due to the absence of disturbance regimes (e.g., grazing, fire) that these species often require [76,77]. Nevertheless, studies suggest that larger-scale connectivity provided by CRP cover in agricultural landscapes may still support positive outcomes for these disturbance-dependent bird species [78]. These examples underscore the importance of considering both spatial scale and management practices—such as mid-contract management techniques that reintroduce disturbance—when evaluating CRP's effectiveness for grassland bird conservation. Additional research that includes other species, both avian as well as non-avian wildlife, and that considers the above-described types of region and species-specific responses to particular management approaches and types of disturbance, can help to bolster conservation efforts and add to the list of CRP success stories.

## Study limitations

While all of the studies in our database examined the effect of CRP, many did not include necessary information to address our objective of characterizing trends in the literature. For example, of the 577 studies only 9% (n = 56) reported the specific CRP conservation practices that were carried out on the enrolled CRP lands in the study. As previously mentioned, early CRP practices primarily consisted of planting prescribed grass mixes (CP1, CP2); therefore, authors of early studies may have felt it was unnecessary to report specific Conservation Practices. It is also possible that details about

the type of CRP implemented in the study areas were confidential and thus not available to researchers even when they had permission to conduct research on CRP lands. Other details like age, size and number of CRP sites were infrequently reported, or were averaged across multiple sites, making it difficult to extract this information precisely. Because the CRP is not one-size-fits-all, reporting of CRP CP codes and other CRP practice details such as vegetative cover that was planted and management practices that were used is important in future studies both for researchers attempting to measure CRP benefits and stakeholders seeking to understand CRP outcomes.

Our database only included studies in which the term "Conservation Reserve Program" was used in a searchable way (i.e., in the title, abstract, or keywords). Because of this we were able to capture some studies that evaluated other USDA programs, such as the Conservation Reserve Enhancement Program (CREP) [79,80]; however, our database is not an exhaustive search of these programs. We also note that our database only includes studies that explicitly evaluated effects of CRP directly. Many other studies that we excluded used CRP land but did not evaluate effects of CRP or conflated the program with other similar land use practices, making it impossible to isolate unique effects of CRP. We also excluded studies not published in peer-reviewed journal articles (e.g., theses, government reports), which may represent a substantial fraction of CRP studies. Although such publications can contain information that is highly useful for conservation and policy activities, this gray literature is not indexed and searchable in the same way as databases of journal articles (e.g., Web of Science), and is therefore more difficult and time consuming to thoroughly capture. Notwithstanding these limitations in data availability, formal meta-analyses of CRP effects and leveraging the wealth and diversity of CRP research that we have highlighted in this scoping review, would provide broader scale understanding of CRP's benefits as well as the moderating factors influencing its benefits. Factors that could be evaluated using meta-analyses include, for example, those related to regional variation, conservation practices used, methods of research employed, and specific endpoints evaluated (e.g., for wildlife and vegetation, the taxonomic groups studied).

## Conclusion

This scoping review of the Conservation Reserve Program literature highlights the significant contributions of CRP to environmental conservation, agricultural productivity, and socioeconomic benefits. The research spans multiple dimensions, including wildlife habitat restoration, environmental quality improvement, and carbon sequestration. The largest number of studies focused on air, soil, and water, while vegetation studies and social surveys were the least represented. Notably, most wildlife research focused on birds, leaving a gap in studies on fish, reptiles, amphibians, and other less-studied taxa. The review also reveals a geographic concentration of research in the Great Plains, underscoring the need for more studies in other ecologically distinct and important regions like the Pacific Northwest. In addition, future research that integrates ecological, agronomic, and socioeconomic outcomes will be especially valuable for informing multi-objective conservation planning. Addressing these gaps, particularly with more long-term studies and a broader range of taxa, especially in the context of changes to how this program is being implemented as well as anthropogenic changes to the environment will be essential to fully understand and enhance the CRP's support for biodiversity and ecosystem services across the U.S.

## Supporting information

**S1 Table. All studies collected in literature search.**
(XLSX)

**S2 Table. CRP acreage and number of CRP studies occurring in each U.S. state.**
(DOCX)

**S3 Table. Taxonomic information and number of studies collected for birds.**
(CSV)

**S1 File. PRISMA-ScR-Checklist.**
(DOCX)

## Acknowledgments

We thank all authors whose work was included in our scoping review. The findings and conclusions in this presentation are those of the author(s) and should not be construed to represent any official USDA or U.S. Government determination or policy.

## Author contributions

**Conceptualization:** Mark P. Nessel, Karen Maguire, Courtney J Duchardt, Scott R. Loss.

**Data curation:** Mark P. Nessel.

**Formal analysis:** Mark P. Nessel.

**Funding acquisition:** Karen Maguire, Courtney J Duchardt, Scott R. Loss.

**Investigation:** Karen Maguire, Rich Iovanna, Courtney J Duchardt, Scott R. Loss.

**Methodology:** Mark P. Nessel.

**Project administration:** Karen Maguire, Courtney J Duchardt, Scott R. Loss.

**Supervision:** Karen Maguire, Courtney J Duchardt, Scott R. Loss.

**Validation:** Mark P. Nessel, Karen Maguire, Rich Iovanna, Courtney J Duchardt, Scott R. Loss.

**Visualization:** Mark P. Nessel.

**Writing – original draft:** Mark P. Nessel, Karen Maguire, Courtney J Duchardt, Scott R. Loss.

**Writing – review & editing:** Mark P. Nessel, Karen Maguire, Rich Iovanna, Courtney J Duchardt, Scott R. Loss.

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
