## [Decision Letter · Decision Letter 0]

2 Apr 2025

PONE-D-24-54508Scoping Review and synthesis of the literature on the outcomes of the Conservation Reserve ProgramPLOS ONE

Dear Dr. Nessel,

Thank you for submitting your manuscript to PLOS ONE. After careful consideration, we feel that it has merit but does not fully meet PLOS ONE’s publication criteria as it currently stands. Therefore, we invite you to submit a revised version of the manuscript that addresses the points raised during the review process.

We look forward to receiving your revised manuscript.

Kind regards,

Marcela Pagano, Ph.D, M.D.

Academic Editor

PLOS ONE

Journal Requirements:

“This research was funded by the U.S. Department of Agriculture (USDA) Natural Resources Conservation Service (Award # NR233A750023C009), the USDA Economic Research Service (Agreement #58-6000-2-0071), and USDA Hatch Grant funds through the Oklahoma Agricultural Experiment Station (OKL03231 and OKL02915). We thank all authors whose work was included in our scoping review. The findings and conclusions in this presentation are those of the author(s) and should not be construed to represent any official USDA or U.S. Government determination or policy. “

“This research was funded by the U.S. Department of Agriculture (USDA) Natural Resources Conservation Service (Award # NR233A750023C009), the USDA Economic Research Service (Agreement #58-6000-2-0071), and USDA Hatch Grant funds through the Oklahoma Agricultural Experiment Station (OKL03231 and OKL02915). We thank all authors whose work was included in our scoping review. The findings and conclusions in this presentation are those of the author(s) and should not be construed to represent any official USDA or U.S. Government determination or policy.”

Reviewers' comments:

Reviewer's Responses to Questions

**Comments to the Author**

1. Is the manuscript technically sound, and do the data support the conclusions?

Reviewer #1: Yes

Reviewer #2: Yes

2. Has the statistical analysis been performed appropriately and rigorously? 

Reviewer #1: No

Reviewer #2: N/A

3. Have the authors made all data underlying the findings in their manuscript fully available?

Reviewer #1: Yes

Reviewer #2: Yes

4. Is the manuscript presented in an intelligible fashion and written in standard English?

Reviewer #1: Yes

Reviewer #2: Yes

5. Review Comments to the Author

Reviewer #1: The authors conducted a systematic review of the effects of Conservation Reserve Program in USA. They systematically searched for studies dealing with Soil/vegetation/wildlife/social and other aspects that were affected by CRP. They also conducted spatial analyses to show which states participating in the program have conducted studies to assess the effects of CRP. The study was done well and summaries have been provided on how many studies were conducted on different habitat/organismal factors affected by CRP. However, the authors did not provide good syntheses in the Discussion. For example, there is good summaries on how many studies covered soil improvement and from which State(s), but there are limited or no syntheses extracted from these studies as discussion points. The same is true of vegetation, wildlife, social studies etc. I feel the paper would improve a lot more if such information is summarized and synthesized from the substantial numbers of papers to enrich the discussion. I suggest major review and i have appended a marked up version of the manuscript where I propose these changes need to be incorporated.

Reviewer #2: Also attached as a PDF:

Greetings,

Thank you for the opportunity to review “Scoping Review and synthesis of the literature on the outcomes of the Conservation Reserve Program” by Nessel et al. I feel this is an interesting manuscript and an important topic; as it does a good job doing the tedious task of sorting through 40 years of literature (mostly focused on the last 25) to contribute to scientific understanding on what/where/and how Conservation Reserve Program (CRP) research is being applied.

Management decisions are guided by regional conservation views and the authors document an extensive literature search/meta-analysis using five major categories— studies of wildlife, vegetation, air/soil/water, productivity, and social aspects to highlight and better understand where additional research on CRP is needed.

Some thoughts/suggestions on the paper:

Throughout the manuscript the authors are inconsistent with their use of a comma after etc.

Line #34: I respect the difficulty separating the categories as many CRP studies are interrelated

Line #48: Why not include the U.S? Or forgo any mention of habitable land around the world all together in favor of U.S. habitable land, as this would be more topical to the paper.

Line #68: I don’t think “Management practices” is correct, these are conservation practices. Either change to “conservation” or just delete “management”.

Line #72: I think if the authors looked hard enough they could find evidence of CRP benefits dating back to the late 1980’s/early 1990’s, but since most of the literature is not electronically available from back then (which the authors do mention), it’s probably not too critical. But it would be interesting to find the first source that champions the benefits of CRP shortly after its establishment in 1985.

Line #99: I am a little shocked that after 40 years this is considered the first scoping review of CRP literature, but I have no evidence to show otherwise. I do, however, believe a large number of papers discuss the importance of continued research on the paper’s subject matter as very rarely does one research paper answer all of the questions. So I’m not sure it is fair to say future research on important aspects have not been emphasized in the literature. Maybe it has not been included in a scoping review, but on a case-by-case basis, I think the majority of discussion sections deal with the need for more research on the subject.

Line #129: Reiterating what I said earlier, it is a bit of a misnomer because a good portion of the papers written in the first 15 years of the CRPs existence may be excluded from your scoping review as some drastic changes happened to the landscape in those early years when tens of millions of acres went from row-crop to grassland. But this is more of a comment then a suggested edit. And I appreciate the authors make mention of the year 2000 as a point of reference (although they do frequently cite pre-2000 papers).

Line #153: Delete practice

Line #194: Average of 20 studies fitting the scoping review criteria, or of 20 studies in general?

Line #282: Was CREP included? I know the CRP is a vast program with many offshoots, but if one were to find a CRP study on amphibians, I would expect to look at CREP studies.

Line #282: I think bats are a bit of a stretch to be researched in CRP grasslands.

Line #322: Similar to amphibians and bats, I don’t think one would find, or expect to find much literature on the impacts of fish on general CRP (if that was the focus) as it is primarily a grassland program. Although one could valiantly argue that fish could be negatively affected by sediment runoff or positively affected by buffering of streams.

Line #425: Yes and no. I agree, that further research is needed, but this has to do with how programmatic changes (e.g., more frequent emergency use) and climatic changes (e.g., more frequent droughts) are affecting the role of CRP. Forty years of literature on the CRP has done an adequate job of explaining direct and indirect benefits born from the program.

Line #453: Good point. Abundance alone can misleading.

Line #484: Be consistent with capitalization. LPC and GPC are inconsistent

Line #525: delete (CRP)

Table S1: This information is old. Update it using this website: Conservation Reserve Program (CRP) Statistics | Farm Service Agency

The CRP has upwards of 40-50 practices, depending how you look at it. My gut assumption was there were many more studies on CRP and this scoping review failed to find them. But the authors clearly define in the methods the sideboards used to narrow the scope and focus. I appreciate the amount of work dedicated to this review, as I am familiar with how labor intensive these efforts can become.

Because of the vastness of the studies reviewed I feel the authors were successful in making every attempt to do a comprehensive review of the literature.

I will confirm though, as evidence of value, that I will incorporate understanding gleaned from the views and impacts resultant from this manuscript into expanding future research into underrepresented CRP lands.

6. PLOS authors have the option to publish the peer review history of their article (what does this mean? ). If published, this will include your full peer review and any attached files.

**Do you want your identity to be public for this peer review?** For information about this choice, including consent withdrawal, please see our Privacy Policy .

Reviewer #1: No

Reviewer #2: No

---

## [Author Response · Author response to Decision Letter 1]

24 Apr 2025

April 24, 2025

Dear Dr. Pagano,

We thank you for giving us the opportunity to respond to the thoughtful remarks of the two reviewers and re-submit our manuscript to PLOS ONE. We find the comments very helpful, fair, and constructive. We have put a considerable effort into fully revising our manuscript and carefully addressed all the concerns and recommendations. We hope you agree that our manuscript has benefited considerably from those improvements and it is now suitable for publication in PLOS ONE.

Regarding Reviewer 1’s comments, most of these asked us to provide additional synthesis of the literature in the form of summaries of the findings of CRP studies within different areas of inquiry (e.g., a synthesis about what the literature says about effects of CRP on soil nutrients, wildlife, etc.). Although we provided thorough responses to all of Reviewer 1’s comments, we decided not to make revisions of this sort because the objective of a scoping review is to characterize trends, emphases, and gaps in what has and has not been studied in a field of research, and not to qualitatively summarize the findings of that literature. We agree that a narrative review or meta-analysis study with the goal of synthesizing the literature in this way would be extremely useful, and indeed we are currently finalizing a meta-analysis paper synthesizing the effects of CRP on wildlife outcomes. However, providing this type of synthesis along with our scoping review results is intractable; to address these comments fully, we would arguably need to double the length of the Discussion section, or even draft a different type of review paper. We hope this rationale for maintaining the scope of the paper as a scoping review, and please note that, to address the reviewer’s concerns, we did add clarification about the intended scope of the paper in the Methods section. Below we provide our detailed responses to the reviewers’ comments. Our replies are in bold font. The line numbers we report in our response correspond to the unmarked version of our revised paper without tracked changes.

We look forward to your assessment of this revised version of the manuscript.

Sincerely, and on behalf of my co-authors,

Mark Nessel

Reviewer #1:

The authors conducted a systematic review of the effects of Conservation Reserve Program in USA. They systematically searched for studies dealing with Soil/vegetation/wildlife/social and other aspects that were affected by CRP. They also conducted spatial analyses to show which states participating in the program have conducted studies to assess the effects of CRP. The study was done well and summaries have been provided on how many studies were conducted on different habitat/organismal factors affected by CRP. However, the authors did not provide good syntheses in the Discussion. For example, there is good summaries on how many studies covered soil improvement and from which State(s), but there are limited or no syntheses extracted from these studies as discussion points. The same is true of vegetation, wildlife, social studies etc. I feel the paper would improve a lot more if such information is summarized and synthesized from the substantial numbers of papers to enrich the discussion. I suggest major review and i have appended a marked up version of the manuscript where I propose these changes need to be incorporated.

We thank the reviewer for their comments. Because this is a scoping review and not a meta-analysis we intentionally did not extract or synthesize the results from our studies. A scoping review isn't a review of what the literature has found but instead is a review of what the literature has and hasn't evaluated. As such, we only report on things we could tally or measure from the papers we extracted, and avoid editorializing or selecting examples that could be misinterpreted by a reader as an overall effect when it is, in fact, a single anecdotal example from our database. As an example, the Grasshopper Sparrow is the most common bird in our database, appearing in 50 studies. If we pick any single study to highlight we might find CRP to have a positive, negative, or non-significant effect, in the specific habitat, and on whichever metric the authors of the studies looked at.

In response to this comment, and in an effort to further emphasize the above point, a sentence was added to the Study categorization section of the methods: “Because this is a scoping review and not a meta-analysis or narrative review, and therefore, because our objective was to characterize trends, emphases, and gaps in what has and has not been studied with regard to CRP effects on wildlife, we intentionally did not extract or synthesize the results from our studies, and instead only focused on what the literature did and did not evaluate.” (See lines 149 to 152). Also, to avoid confusion about the scope of the paper, we deleted “synthesis” from the title and from other locations within the paper.

Line 69: Spell out what CP (Conservation Practice) means for international audiences when first mentioned.

We have expanded the sentence to spell out what these practices are. The sentence now reads: “These practices, which are indicated by “CP” followed by a number, include "Establishment of Permanent Native Grasses" (CP2), "Riparian Buffers" (CP22), and "Habitat Buffers for Upland Birds" (CP33) (USDA-FSA 2024).”

Line 389: A broader synthesis on how soil, water and other aspects of habitat quality were improved is needed. Which soil parameters were improved (N? P? pH?) ? Which air quality measures were improved? The social aspects should be separated out and a synthesis needs to be provided on broadly which social aspects were improved.

As described in greater detail above, synthesizing the results of the studies collected is outside the scope of this type of review paper.

Line 427: I would have wanted more discussion/synthesis on the effects of the CRP on wildlife. Specific examples such as "species A increased due the program in the State X" or "community diversity or overall biomass of a species increased in riparian habitats"... this would make the discussion more informative compared to whether 'n' number of studies addressed issues relating to biomass and diversity.

As described in greater detail above, synthesizing the results of the studies collected is outside the scope of this type of review paper.

Line 432: Ok. But what do some of these short term studies driven by graduate students demonstrate. Some synthesis would be essential

Thank you for this comment. In order to clarify what types of conclusions are possible with short term research we have added the sentence: “Although short-term studies can provide invaluable insights about associations between CRP and outcomes like wildlife and vegetation abundance, occurrence, species diversity, and reproductive rates, long-term and large-scale studies are important for accurately measuring ecological trends and capturing lag effects that might not manifest until years after management is applied.” (See lines 437 to 441).

Line 458: So, make this more informative by picking some examples of species that were increased due to either land management acitivies but was mis-interpretted as a those associated with CRP.

This is another similar comment about synthesis as those above, and please see our above, more-detailed responses.

Line 464: I agree that more emphasis needs to be placed on other taxa. However, what do some of the 68% of biased studies show? Are some of the non-native species primarily benefitted or are some of the common species increased? Please provide specific examples and sythesize the information about overall conclusions of trends made by the authors of the study.

Again, see the above more-detailed responses regarding literature syntheses; providing all of this requested information would greatly increase the length of the discussion, and arguably would need to be done in an entirely separate narrative review paper.

Line 480: For example? which non-listed species of the ACAD watch list were considered?

Line 485: Ok

We don’t believe this comment requires a specific revision. After the text highlighted by the comment, we continue to list bird species collected in our studies as well as report all of the bird species from our collected studies in Table S3. We interpret the “OK” left by the reviewer a few lines down, as them agreeing with this interpretation.

Line 485: What is the overall nature of the benefit to these species: did their densities increase? Did they have higher reproductive success?

Synthesizing the results of the studies collected is outside the scope of this review.

Reviewer #2:

Greetings,

Thank you for the opportunity to review “Scoping Review and synthesis of the literature on the outcomes of the Conservation Reserve Program” by Nessel et al. I feel this is an interesting manuscript and an important topic; as it does a good job doing the tedious task of sorting through 40 years of literature (mostly focused on the last 25) to contribute to scientific understanding on what/where/and how Conservation Reserve Program (CRP) research is being applied.

Management decisions are guided by regional conservation views and the authors document an extensive literature search/meta-analysis using five major categories— studies of wildlife, vegetation, air/soil/water, productivity, and social aspects to highlight and better understand where additional research on CRP is needed.

Some thoughts/suggestions on the paper:

Throughout the manuscript the authors are inconsistent with their use of a comma after etc.

Because "etc." was only used once in the manuscript, we were unclear about how to address this comment. We reviewed comma use after other abbreviations (e.g.) and tried to maintain consistency, but would be happy to further address this issue with additional clarification

Line #34: I respect the difficulty separating the categories as many CRP studies are interrelated

Thank you for this comment, which we do not interpret as requesting a specific revision.

Line #48: Why not include the U.S? Or forgo any mention of habitable land around the world all together in favor of U.S. habitable land, as this would be more topical to the paper.

We have added information about US agricultural land to the statement. It now reads: “The Food and Agriculture Organization of the United Nations estimates that agricultural land makes up 48 million km2, or about 45% of all habitable land on earth (FAO 2024; Ritchie & Roser 2019), and in 2022, 45.1% of U.S. land area was estimated to be devoted to agricultural use (World Bank 2025).” (See lines 44 to 48)

Line #68: I don’t think “Management practices” is correct, these are conservation practices. Either change to “conservation” or just delete “management”.

We have removed management from the sentence. It now reads: “Sharing a common theme of perennial cover, the broad range of practices…” (See lines 64 to 65)

Line #72: I think if the authors looked hard enough they could find evidence of CRP benefits dating back to the late 1980’s/early 1990’s, but since most of the literature is not electronically available from back then (which the authors do mention), it’s probably not too critical. But it would be interesting to find the first source that champions the benefits of CRP shortly after its establishment in 1985.

Thank you for this comment, which we do not interpret as requesting a specific revision.

Line #99: I am a little shocked that after 40 years this is considered the first scoping review of CRP literature, but I have no evidence to show otherwise. I do, however, believe a large number of papers discuss the importance of continued research on the paper’s subject matter as very rarely does one research paper answer all of the questions. So I’m not sure it is fair to say future research on important aspects have not been emphasized in the literature. Maybe it has not been included in a scoping review, but on a case-by-case basis, I think the majority of discussion sections deal with the need for more research on the subject.

Thanks for catching us on perhaps overstepping with our statement of novelty of the research. We have changed the language of this sentence in order to not overstate the novelty. The sentence now reads: “With this review, we seek to inform and guide future research on important aspects of CRP that have to this point not received formal empirical evaluations in the literature.”

Line #129: Reiterating what I said earlier, it is a bit of a misnomer because a good portion of the papers written in the first 15 years of the CRPs existence may be excluded from your scoping review as some drastic changes happened to the landscape in those early years when tens of millions of acres went from row-crop to grassland. But this is more of a comment then a suggested edit. And I appreciate the authors make mention of the year 2000 as a point of reference (although they do frequently cite pre-2000 papers).

Thank you for this comment, which we do not interpret as requesting a specific revision.

Line #153: Delete practice

All text that said “CP Practices” has been changed to “Conservation Practices”

Line #194: Average of 20 studies fitting the scoping review criteria, or of 20 studies in general?

Text was added to this sentence to clarify the statement. The new sentence reads: “During the 30-year period between 1994 and 2023 an average of 20 studies examining the effects of CRP were published per year ranging from 11 studies in 1995 to 36 studies in 2016.”

Line #282: Was CREP included? I know the CRP is a vast program with many offshoots, but if one were to find a CRP study on amphibians, I would expect to look at CREP studies.

We did not explicitly search for CREP (see search terms in the methods), but CREP studies were captured in our search, and included in our database. We had already acknowledged in the Study Limitations section of the discussion that there are some potential problems that might arise when trying to collect every paper of interest; however, in response to this comment we also revised to mention that our review seemed to also capture studies on very closely related offshoot programs, like CREP: “Because of this we were able to capture some studies that evaluated sub-programs, such as the Conservation Reserve Enhancement Program (CREP) (e.g., Manley & Mathias 2017; Wentworth et al. 2010); however, our database is not an exhaustive search of these offshoots” (See lines 514 to 517)

Line #282: I think bats are a bit of a stretch to be researched in CRP grasslands.

Our focus was not only on CRP grasslands, but all land enrolled in the CRP program. Further, bats occur in airspace over grasslands and are a species of interest in many agricultural studies. We believe it worth pointing out that this important species was totally absent from the CRP studies we collected.

Line #322: Similar to amphibians and bats, I don’t think one would find, or expect to find much literature on the impacts of fish on general CRP (if that was the focus) as it is primarily a grassland program. Although one could valiantly argue that fish could be negatively affected by sediment runoff or positively affected by buffering of streams.

In the same way as bats, fish occupy freshwater bodies adjacent to CRP land, and are the subject of many studies examining the effect of agricultural practices on their ecology. We think it’s worthy to highlight their complete absence from the CRP literature that we were able to collect, even if we don’t expect them to be the preponderance of the study taxa.

Line #425: Yes and no. I agree, that further research is needed, but this has to do with how programmatic changes (e.g., more frequent emergency use) and climatic changes (e.g., more frequent droughts) are affecting the role of CRP. Forty years of literature on the CRP has done an adequate job of explaining direct and indirect benefits born from the program.

Text was added to the sentence to clarify our statement: “Further research that continues to evaluate the wildlife and vegetation-mediated ecosystem services provided by CRP, especially how these services are affected by implement

---

## [Decision Letter · Decision Letter 1]

27 May 2025

PONE-D-24-54508R1Scoping Review of the literature on outcomes of the Conservation Reserve Program.PLOS ONE

Dear Dr. Nessel,

Thank you for submitting your manuscript to PLOS ONE. After careful consideration, we feel that it has merit but does not fully meet PLOS ONE’s publication criteria as it currently stands. Therefore, we invite you to submit a revised version of the manuscript that addresses the points raised during the review process.

We look forward to receiving your revised manuscript.

Kind regards,

Marcela Pagano, Ph.D, M.D.

Academic Editor

PLOS ONE

Reviewers' comments:

Reviewer's Responses to Questions

**Comments to the Author**

1. If the authors have adequately addressed your comments raised in a previous round of review and you feel that this manuscript is now acceptable for publication, you may indicate that here to bypass the “Comments to the Author” section, enter your conflict of interest statement in the “Confidential to Editor” section, and submit your "Accept" recommendation.

Reviewer #3: (No Response)

2. Is the manuscript technically sound, and do the data support the conclusions?

Reviewer #3: Partly

3. Has the statistical analysis been performed appropriately and rigorously? 

Reviewer #3: N/A

4. Have the authors made all data underlying the findings in their manuscript fully available?

Reviewer #3: Yes

5. Is the manuscript presented in an intelligible fashion and written in standard English?

Reviewer #3: Yes

6. Review Comments to the Author

Reviewer #3: Overall, understanding research emphases and gaps in the Conservation Reserve Program (CRP) literature is an important question for continental conservation of cropland modified ecosystems. The article presents a thorough scoping review of the primary literature for research on the CRP. The objectives of the study were to 1) characterize spatial and temporal trends in CRP research, 2) categorize CRP research by type, including studies of wildlife; vegetation; air, soil, and water; and more human-focused aspects, and 3) evaluate detailed characteristics of studies that focus on responses of wildlife and vegetation to CRP. The findings indicate most studies are concentrated in the Great Plains, leaving regions such as the Pacific Northwest underrepresented. For wildlife-related outcomes of CRP, birds dominated the literature, while research on other taxa such as fish, reptiles, and amphibians are sparse.

The main considerations identified for improving the manuscript include 1) clarifying concepts and terminology for wildlife habitat relationships, 2) revising claims about lack of temporal trends not supported by data, 3) providing a caveat or discussion on the sensitivity of conclusions to double counting frequency of studies, and 4) discussing the importance of disturbance and management for species-specific habitat suitability of CRP.

Specific comments

Lines 43 – 49: Although it is appropriate to discuss agriculture in general to introduce the study, agriculture can refer to animal production, forestry, soil science, etc. Consider framing study around conservation problems associated with farming and cultivated cropland, resulting in grassland loss and associated habitat loss and fragmentation for grassland species.

Lines 54 – 56: Consider mentioning Farm Bill Legislation as the funding mechanism.

Lines 100 – 185: No quantities to be summarized or statistical methods were described making it difficult to understand how data from the literature review will be used to address the study objectives. Consider adding a sentence or two to describe the summaries used to draw conclusions from the review.

Lines 141 – 143: Consider adding a sentence to indicate that summaries were calculated with some studies counted more than once. This disclaimer appears in the figure captions, but the issue of double counting studies in the summaries must be stated in the article. This may be yanking around results based on summarizing the frequency of studies. Even a scoping review requires an evaluation that claims are supported by the data.

Line 145: Consider a revision to improve precision of the concept for “wildlife habitat” used throughout the article. Because habitat is a species-specific concept, the concept of general “wildlife habitat” does not exist in wildlife ecology (Hall et al. 1997, Johnson 2007). In a scoping review, it is important to clarify concepts for the research area, and use of a term such as “wildlife habitat” incorrectly implies that CRP functions equally as habitat across species when habitat requirements for CRP vegetation is specific and unique for individual species.

Table 1: Was species richness used as an indication of diversity? If so, consider adding species richness to the diversity examples item.

Table 1: Abundance is listed as a sperate row, but density, a measurement of abundance, is listed under Distribution. Consider adding a rational for having the same metric listed on different rows.

Figure 1: Consider adding figure axis titles, and fonts large enough to be read. Consider removing incomplete data for year 2024 from the figure because it is not comparable to the other time occasions.

Figure 2: Consider adding axis titles, and fonts large enough to be read. Consider labelling the bars with the numbers of studies so that the readers can decide if there is a trend in the number of studies over time.

Figure 4: Consider making the headings large enough to be read

Figure 5: Consider adding axis titles, and fonts large enough to be read

Lines 215 – 216: Consider quantifying in terms of a proportion or percentage

Lines 218 – 219: Consider providing a correlation for the relationship

Line 251: See above, questionable use of terms “wildlife habitat” and “habitat quality”

Figure 7: The figure is out of sequence. The figure needs axis titles, and fonts large enough to be read.

Lines 319 – 320: Consider a revision to justify this sentence. Figure 2 indicates the number of studies increased over time.

Lines 329 – 330: How was this determined? Consider discussing the apparent increase in the number of studies shown in Figure 2. A simple glm of apparent frequencies from Fig. 2, shows a large increase in the number of studies over time, and more evidence for a non-linear threshold increase over time

Lines 330 – 332: The annual trend in the proportional change between study types is a different question than over all change over time. Consider providing some evidence for “relative consistent number of studies. This requires some sort of statistical support. For example, one could take the proportion for each type, and then test for correlation with year. That would probably show the same as the over all increase for Air/Soil/Water, Vegetation, and Wildlife, but possibly no increase over time for Productivity or Social.

Figure 6: The figure needs axis titles, and fonts large enough to be read

Lines 352 – 357: Run-on sentence. Consider discussing the primary reason for studying CRP in regions with high cultivated cropland is to understand the role of CRP to address habitat loss from grassland conversion, something not touched on in the introduction.

Line 361 – 365: Consider adding a sentence to discuss lack of disturbance as the reason why CRP may not provide habitat for the Grasshopper Sparrow in the eastern part of its range. In addition, the disturbance gradient for Grasshopper Sparrow also occurs within the Great Plains, and yet CRP is still an important conservation strategy for addressing habitat loss from grassland conversion for this species.

Lines 371 – 382: Consider conservation goals to balance crop yields with the other CRP biophysical and biodiversity goals (Butsic et al. 2020).

Line 396: See above, no such thing as “wildlife habitat”

Line 413: Consider revising this sentence or the previous conclusions lines 319 – 320 for no trends in the data over time

Line 416 – 417: Consider rephrasing this sentence, habitat is a species-specific concept, CRP does not generally support wildlife habitat, each species has a different habitat relationship with CRP vegetation features.

Line 445: See above, there is no such thing as wildlife habitat.

Lines 482 – 496: Consider discussing the importance of disturbance for many species of grassland birds (Derner et al. 2009), and the potentially the importance of CRP different spatial scales.

Lines 493 – 494: Consider providing relevant references for Chestnut collared Longspur and Baird's Sparrow, or choose other species to discuss. The references cited are inappropriate for habitat relationships of these species on the above list. There is actually a fair bit of controversy on the importance of CRP to local habitat requirements for species such as Chestnut collared Longspur and Baird's Sparrow, and well as disturbance-dependent species (Derner et al. 2009). Many sources, including Shaffer et al. (2020), suggest most CRP provides poor habitat for Baird's Sparrow. The Chestnut collared Longspur often requires habitat conditions with greater disturbance than is afforded by CRP. Nevertheless, Pavlacky et al. (2022) found positive effects for Chestnut collared Longspur and Baird's Sparrow from large-scale connectivity of increasing CRP cover in the surrounding landscape.

This may also be a good place to discuss the importance of mid-contract management CRP Grasslands to introduce disturbance and provide suitable habitat for disturbance-dependent species.

Lines 498 – 532: Consider short paragraph to discuss the sensitivity of the conclusions to double counting studies in the summaries presented in the results section. This may be yanking around the conclusions based on summarizing the frequency of studies. Even a scoping review requires an evaluation that claims are supported by the data.

Lines 534 – 547: Consider discussing the importance of addressing information gaps to achieve multi-objective conservation planning to balance transdisciplinary outcomes.

Line 537: See above, no such thing as wildlife habitat

Literature Cited

Butsic, V., T. Kuemmerle, L. Pallud, K. J. Helmstedt, L. Macchi, and M. D. Potts. 2020. Aligning biodiversity conservation and agricultural production in heterogeneous landscapes. Ecological Applications 30:e02057.

Derner, J. D., W. K. Lauenroth, P. Stapp, and D. J. Augustine. 2009. Livestock as ecosystem engineers for grassland bird habitat in the western Great Plains of North America. Rangeland Ecology & Management 62:111-118.

Hall, L. S., P. R. Krausman, and M. L. Morrison. 1997. The habitat concept and a plea for standard terminology. Wildlife Society Bulletin 25:173-182.

Johnson, M. D. 2007. Measuring habitat quality: a review. The Condor 109:489-504.

Pavlacky, D. C., Jr., A. W. Green, T. L. George, R. Iovanna, A. M. Bartuszevige, M. D. Correll, A. O. Panjabi, and T. B. Ryder. 2022. Landscape-scale conservation mitigates the biodiversity loss of grassland birds. Ecological Applications 32:e2548.

Shaffer, J. A., L. D. Igl, D. H. Johnson, M. L. Sondreal, C. M. Goldade, M. P. Nenneman, and B. R. Euliss. 2020. The effects of management practices on grassland birds: Baird’s sparrow (Centronyx bairdii). Professional Paper 1842–HH. in D. H. Johnson, L. D. Igl, J. A. Shaffer, and J. P. DeLong, editors. The effects of management practices on grassland birds. Professional Paper 1842. United States Department of the Interior, U. S. Geological Survey, Reston, Virginia, USA. <. Accessed 21 March 2022.

7. PLOS authors have the option to publish the peer review history of their article (what does this mean? ). If published, this will include your full peer review and any attached files.

**Do you want your identity to be public for this peer review?** For information about this choice, including consent withdrawal, please see our Privacy Policy .

Reviewer #3: **Yes: ** David C. Pavlacky Jr.

---

## [Author Response · Author response to Decision Letter 2]

8 Jul 2025

June 20, 2025

Dear Dr. Pagano,

We thank you for the opportunity to submit this second revised version of our manuscript to PLOS ONE. We appreciate the continued engagement from the reviewers and editor, and we have carefully addressed the additional comments raised during this round of review.

As a reminder, this revision follows a previous set of reviewer comments, which we addressed in full in our initial resubmission, and received no further feedback. We understand that this current round reflects further input from an additional reviewer, and we have treated these as a distinct set of revisions.

In this revision, we made several modest changes to the manuscript to improve clarity and accuracy. These include: clarifying our interpretation of temporal patterns in the literature; refining figure formatting and labeling (including annual publication counts and font adjustments); and incorporating small textual clarifications to address reviewer concerns around terminology and interpretation.

We hope these changes further strengthen the manuscript and demonstrate our responsiveness to the feedback. Below we provide our detailed responses to the reviewers’ comments. Our replies are in bold font. The line numbers we report in our response correspond to the unmarked version of our revised paper without tracked changes.

Sincerely, and on behalf of my co-authors,

Mark Nessel

Reviewer #3: Overall, understanding research emphases and gaps in the Conservation Reserve Program (CRP) literature is an important question for continental conservation of cropland modified ecosystems. The article presents a thorough scoping review of the primary literature for research on the CRP. The objectives of the study were to 1) characterize spatial and temporal trends in CRP research, 2) categorize CRP research by type, including studies of wildlife; vegetation; air, soil, and water; and more human-focused aspects, and 3) evaluate detailed characteristics of studies that focus on responses of wildlife and vegetation to CRP. The findings indicate most studies are concentrated in the Great Plains, leaving regions such as the Pacific Northwest underrepresented. For wildlife-related outcomes of CRP, birds dominated the literature, while research on other taxa such as fish, reptiles, and amphibians are sparse.

The main considerations identified for improving the manuscript include 1) clarifying concepts and terminology for wildlife habitat relationships, 2) revising claims about lack of temporal trends not supported by data, 3) providing a caveat or discussion on the sensitivity of conclusions to double counting frequency of studies, and 4) discussing the importance of disturbance and management for species-specific habitat suitability of CRP.

Thank you for these constructive comments and the positive appraisal. We respond to all of these major considerations below.

Specific comments

Lines 43 – 49: Although it is appropriate to discuss agriculture in general to introduce the study, agriculture can refer to animal production, forestry, soil science, etc. Consider framing study around conservation problems associated with farming and cultivated cropland, resulting in grassland loss and associated habitat loss and fragmentation for grassland species.

Thank you for this suggestion. In order to focus on habitat loss associated with farming we changed the sentence to read “In particular, land use changes associated with agricultural crop production are contributing substantially to biodiversity loss (Caro et al. 2022; Jaureguiberry et al. 2022).” [lines 47-48]

Lines 54 – 56: Consider mentioning Farm Bill Legislation as the funding mechanism.

We added a clause to the sentence indicating the Farm Bill administers the CRP program.

Lines 100 – 185: No quantities to be summarized or statistical methods were described making it difficult to understand how data from the literature review will be used to address the study objectives. Consider adding a sentence or two to describe the summaries used to draw conclusions from the review.

In order to clarify our approach for generating descriptive summaries in support of addressing our objectives, we added a sentence to the methods: “To address the objectives of this scoping review, we generated total counts of studies and calculated proportions of studies in different categories in order to identify trends and gaps in the literature.” [lines 152-154]

Lines 141 – 143: Consider adding a sentence to indicate that summaries were calculated with some studies counted more than once. This disclaimer appears in the figure captions, but the issue of double counting studies in the summaries must be stated in the article. This may be yanking around results based on summarizing the frequency of studies. Even a scoping review requires an evaluation that claims are supported by the data.

We added some text to further clarify how we counted studies that belonged under multiple categories: “As with the study categories above, many of these categories were not mutually exclusive, meaning one study could contribute multiple data points (e.g., a study that evaluated multiple conservation practices). However, because such studies were only included in descriptive summaries of total counts of studies or to calculate proportions of total studies (e.g., X% of studies examining bird species),any individual study was only counted once for any single metric that we present” [lines 161-166]

Line 145: Consider a revision to improve precision of the concept for “wildlife habitat” used throughout the article. Because habitat is a species-specific concept, the concept of general “wildlife habitat” does not exist in wildlife ecology (Hall et al. 1997, Johnson 2007). In a scoping review, it is important to clarify concepts for the research area, and use of a term such as “wildlife habitat” incorrectly implies that CRP functions equally as habitat across species when habitat requirements for CRP vegetation is specific and unique for individual species.

We appreciate the reviewer’s point and agree that “habitat” is most accurately defined in species-specific terms, as emphasized by the cited literature. We recognize the risk of overgeneralizing this term in ecological writing and have carefully reviewed our manuscript with this in mind.

As a scoping review, our goal is to synthesize and describe the findings of individual studies, each of which typically defines and evaluates habitat at the species level. When we refer to “wildlife habitat,” we are not suggesting a universal or undifferentiated habitat across all species, but rather using a shorthand to refer to the various species-specific habitats that individual studies have identified in relation to CRP. Our synthesis encompasses these distinct definitions rather than collapsing them into a single concept.

To clarify this in the manuscript and address the concern:

We have revised the term “wildlife habitat” where appropriate to provide greater precision or contextual clarity.

At the first mention of “wildlife habitat” in the Introduction (Line 71), we have added a parenthetical note acknowledging the species-specific nature of habitat and clarifying our intended use of the term in the paper. The revised sentence reads:

“These benefits include providing wildlife habitat (acknowledging that habitat is a species-specific concept, we use ‘wildlife habitat’ as a shorthand for the variety of species-specific habitats reported in the literature)” [lines 71-73]

We hope this clarification and the accompanying edits throughout the manuscript appropriately address the concern while preserving clarity and consistency in the narrative.

Table 1: Was species richness used as an indication of diversity? If so, consider adding species richness to the diversity examples item.

Species richness was included in our Diversity group. It has been added to the examples in Table 1 to indicate this.

Table 1: Abundance is listed as a sperate row, but density, a measurement of abundance, is listed under Distribution. Consider adding a rational for having the same metric listed on different rows.

Density is counted in the Abundance group. We use the word density in the description of the Distribution group. In order to alleviate this confusion the text was changed in the Distribution description to not include the word density.

Figure 1: Consider adding figure axis titles, and fonts large enough to be read. Consider removing incomplete data for year 2024 from the figure because it is not comparable to the other time occasions.

We appreciate the suggestion. We have increased the font size in all bar graphs (Figures 1, 2, 5, 6, and 7) to improve legibility. We also added consistent axis labels across all bar graphs. We believe the reference to 2024 belongs to Figure 2 (which displays numbers of studies for different publication years), not Figure 1. Accordingly, we removed the 2024 data from Figure 2 and clarified the presence of two 2024 studies in the figure legend instead. We also note that font scaling often depends on figure sizing during publication and may display differently depending on screen size and output resolution. Previous reviewers did not raise font legibility as an issue, but we are happy to work with the editor during copyediting to ensure the figures display as clearly as possible.

Figure 2: Consider adding axis titles, and fonts large enough to be read. Consider labelling the bars with the numbers of studies so that the readers can decide if there is a trend in the number of studies over time.

We updated all bar graphs to use a consistent y-axis title ("Number of Studies"). We retained the omission of x-axis titles since the category or year labels are already shown on the axis. We also added numeric labels above each bar in Figure 2 showing the total number of studies published each year.

Figure 4: Consider making the headings large enough to be read

Thank you. As noted above, we increased the font size for all bar graphs to improve clarity.

Figure 5: Consider adding axis titles, and fonts large enough to be read

We increased the font size of bar labels and axis text in all bar graphs and standardized the y-axis title as "Number of Studies," as described above.

Lines 215 – 216: Consider quantifying in terms of a proportion or percentage

We added text to clarify the exact states along to the 98th meridian and report the exact number of studies conducted there: “The largest number of studies have occurred in states in the central part of the U.S.; specifically, 46% (n =230) of studies were in states along the 98th meridian (i.e., North Dakota, South Dakota, Nebraska, Kansas, Oklahoma and Texas), with most of the studies in our dataset occurring in Great Plains states (Figure 3).” [lines 226-229]

Lines 218 – 219: Consider providing a correlation for the relationship

We appreciate the reviewer’s suggestion to provide a correlation between CRP enrollment and the number of studies conducted in each state. However, after consideration, we opted not to conduct a formal correlation analysis for a few reasons. First, CRP enrollment is a dynamic variable that changes annually, while publication dates of CRP studies do not necessarily align with the specific year(s) of CRP data reported in those studies. As a result, a direct correlation between CRP acreage and study count across 50 states using a single year of enrollment data would provide limited interpretability.

To address this issue in a consistent and transparent manner, we used enrollment data from 2017, the year following the peak in study publication, as a fixed reference point. We then provided a quantitative summary: the 10 states with the most CRP acreage in 2017 accounted for 64% of studies with state-level data, averaging 54 studies per state. We also explicitly discussed deviations from this trend (e.g., Washington had fewer studies than expected given its acreage, while Wisconsin and Michigan had more).

We believe this descriptive approach provides meaningful context for understanding spatial research trends without overstating the strength or consistency of the relationship, and we have clarified this rationale in the manuscript.

Line 251: See above, questionable use of terms “wildlife habitat” and “habitat quality”

See above discussion about the use of the term “wildlife habitat”

Figure 7: The figure is out of sequence. The figure needs axis titles, and fonts large enough to be read.

Thank you for this comment. We reviewed the manuscript and confirmed that Figure 7 is cited in sequence following Figure 6. It is possible this comment was related to figure formatting during the review process. Nevertheless, we have double-checked the figure order to ensure accuracy. We also addressed the accompanying comments by updating axis titles and improving font sizes for clarity.

Lines 319 – 320: Consider a revision to justify this sentence. Figure 2 indicates the number of studies increased over time.

We appreciate the reviewer’s suggestion and agree that the original sentence could be misinterpreted as a statistical inference rather than a coarse summary of the literature timeline. To address this, we have revised the sentence to avoid implying a formal test of temporal variation and instead present a descriptive summary of the publication record. The updated text reads: “The published research on CRP has been relatively consistent temporally despite declines in enrolment” [lines 332-333]

Lines 329 – 330: How was this determined? Consider discussing the apparent increase in the number of studies shown in Figure 2. A simple glm of apparent frequencies from Fig. 2, shows a large increase in the number of studies over time, and more evidence for a non-linear threshold increase over time

Thank you for this observation. We agree that a formal statistical model could be used to assess trends in publication frequency over time. However, as our scoping review was not designed to evaluate statistical trends, we avoid quantitative inferences and instead describe the pattern as it appears in the updated version of Figure 2, which now includes annual (rather than binned) counts. To further clarify our intent, we revised the paragraph to remove references to “consistency”: “There were no major year-over-year changes in the type or number of studies published about CRP since the program’s inception in 1985. Beginning in 1994 we observed no multi-year trends in the number of published studies in all 5 categories.” [lines 342-344]

Lines 330 – 332: The annual trend in the proportional change between study types is a different question than over all change over time. Consider providing some evidence for “relative consistent number of studies. This requires some sort of statistical support. For example, one could take the proportion for each type, and then test for correlation with year. That would probably show the same as the over all increase for Air/Soil/Water, Vegetation, and Wildlife, but possibly no increase over time for Productivity or Social.

We agree that the distinction between the total number of studies and the proportional distribution across study types is important. In our analysis, we focused on describing the absolute number of studies per category and how those varied across time, rather than testing whether proportions shifted statistically. This is especially important as some studies would count under multiple categories making interpretation difficult. We have clarified this distinction in the text and avoided using “relatively consistent” to describe trends without statistical support.

Figure 6: The figure needs axis titles, and fonts large enough to be read

As with the other bar graphs, we increased font sizes and added axis titles to Figure 6.

Lines 352 – 357: Run-on sentence. Consider discussing the primary reason for studying CRP in regions with high cultivated cropland is to understand the role of CRP to address habitat loss from grassland conversion, something not touched on in the introduction.

Thank you for this helpful suggestion. We revised the sentence to improve clarity and added a statement acknowledging the disproportionate importance of CRP in regions with si

---

## [Editor Report · Decision Letter 2]

24 Jul 2025

Scoping Review of the literature on outcomes of the Conservation Reserve Program.

PONE-D-24-54508R2

Dear Dr. ulrika Ervander,

We’re pleased to inform you that your manuscript has been judged scientifically suitable for publication and will be formally accepted for publication once it meets all outstanding technical requirements.

Kind regards,

Marcela Pagano, Ph.D, M.D.

Academic Editor

PLOS ONE